# CAUSALSTEWARD: AN AGENTIC DIVIDE-CONQUER-COMBINE COPILOT FOR CAUSAL DISCOVERY

## ABSTRACT

Learning causal models from high-dimensional data is a significant challenge, particularly in real-world settings where violations of core assumptions lead to causal identifiability issues. Although massive amounts of prior knowledge are available, and contain valuable causal information, effectively integrating this knowledge into the causal discovery process remains an open problem. We introduce **C**ausal**ST**eward (CAST), a novel human-in-the-loop framework for interactively assembling large causal models. CausalSteward is a multi-agent collaborative system that tackles high-dimensional causality through a divide-and-conquer approach where large clusters of variables are iteratively partitioned and then separately analyzed. Our framework fuses prior knowledge with a data-driven approach by using tailored tools such as retrieval augmented generation and conditional independence tests. Finally, we use this work to examine the capabilities and limitations of causal reasoning in multi-agent frameworks, and how the human-in-the-loop can contribute to accurate and trustworthy results.

## 1 INTRODUCTION

Modern science and engineering increasingly demand for causal models (Pearl, 2009) that move beyond purely statistical correlation to predict the outcomes of manipulations on the system (interventions) and reasoning about hypothetical scenarios (counterfactuals). Performing such inferences often requires a causal graph representing the network of cause-and-effect relationships, however the true causal graph is rarely known. Existing approaches (Spirtes et al., 2001; Shimizu et al., 2011; Spirtes, 2001) commonly attempt to learn this graph from data by relying on causal discovery methods. However, causal discovery from observational data is fundamentally limited by the assumptions required for causal identifiability (Shimizu et al., 2011; Zhang & Hyvarinen, 2012; Jaber et al., 2018). Violations of these assumptions can result in multiple causal graphs being statistically undistinguishable, which leads to non-identifiable causal relationships and limits accurate causal discovery. These limitations can be overcome by integrating prior knowledge (O'Donnell et al., 2006). Domain experts can manually incorporate their knowledge by imposing edge constraints, but this has limited utility in high-dimensional settings (Constantinou et al., 2023). In particular, unstructured text is a rich source of prior knowledge that is dense of causal information, while being massively available in the web, documents, and more. Recent works using Large Language Models (LLMs) show a promising outlook for automating the extraction of causal information (Ban et al., 2023; Antonucci et al., 2023; Sheth et al., 2025), offering a great tool for the construction of large causal graphs (Abdulaal et al., 2024; Chen et al., 2023). Furthermore, a Human-in-the-Loop (HITL) approach enables domain experts to guide the LLM reasoning process and obtain accurate solutions without requiring tedious manual specification. In this paper we introduce **C**ausal**ST**eward (CAST), a multi-agent causal copilot that discovers large causal graphs by combining human interaction, retrieval augmented generation (RAG), and data-driven methods. Our approach is *human-first*, and CAST has the duty of assisting the human in building a causal model while automating the collection of prior knowledge. CAST adopts a Divide-and-Conquer approach

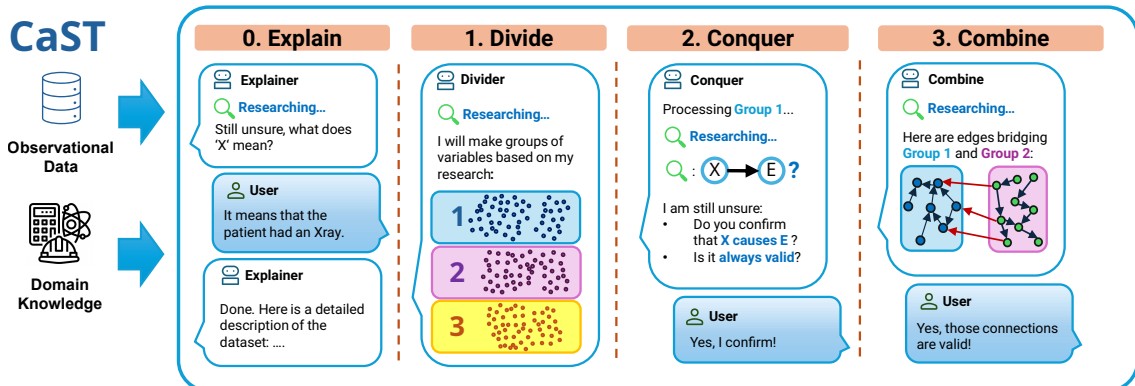

Figure 1: **CausalSteward (CAST)**: This paper develops and showcases the utility of a human-in-the-loop, multi-agent system for causal discovery, leveraging the combined strength of automated discovery from data and prior knowledge of LLM, together with human-expert feedback to incorporate domain specific knowledge.

where agents iteratively partition the variable set based on their likelihood of being causally connected until the right level of granularity is reached. Successively, CAST estimates a local causal graph for each partition based on prior knowledge (retrieved from HITL and RAG), and observational data. Finally, the local causal graphs are merged together, obtaining the global causal graph. We apply CAST to a diverse set of challenging datasets in the domains of manufacturing, neuropathic pain and the CausalChambers benchmark (Gamella et al., 2024). Our experiments evaluate CAST with LLMs of varying capabilities and compare against a set of seven other LLM- and data-driven baselines. Our ablations, which disable the RAG and human-in-the-loop components, demonstrate that only the joint use of both components leads to the strong performance of CAST.

**Contribution:** In this paper we present CAST, a Human-in-the-Loop (HITL) framework for joint collaborative causal discovery integrating tabular data, LLM prior knowledge and RAG search, together with human-feedback back domain specific questions. To the best of our knowledge, we are first to present an interactive multi-agentic framework with a Divide-and-Conquer paradigm that has the ability to discover causal graphs in high-dimensional causal settings.

## 2 PRELIMINARIES AND RELATED WORK

**Notation:** We denote sets with bold $\mathbf{V}$ and variables in lowercase $v_i$. **Causal Models:** Within the framework of Pearlian causality (Pearl, 2009), the most common class of causal models are *Structural Causal Models*, which we define as a 4-tuple $\mathcal{M} := (\mathbf{U}, \mathbf{V}, P_{\mathbf{U}}, \mathcal{F})$ where $\mathbf{U}$ is the set of exogenous variables, $\mathbf{V}$ is the set of endogenous variables, $P_{\mathbf{U}}$ is the probability density function of the exogenous variables $\mathbf{U}$, and $\mathcal{F} = \{f_1, f_2, \ldots, f_n\}$ is the set of *Structural Equations*, where each element is a mapping such that $f_i : u_i \cup Pa_i \to \mathbf{V}_i$, with $u_i \subseteq \mathbf{U}$ and $v_i \subseteq \mathbf{V}$. Every endogenous variable $v_i \in \mathbf{V}$ is coupled to a structural equation that determines its values, i.e. $v_i = f_i(u_i, Pa_i)$.

Structural equations induce a set of dependencies between endogenous variables. Those dependencies can be organized in a directed acyclic graph (DAG) $\mathcal{G}(\mathbf{V}, \mathbf{E})$, where $\mathbf{V}$ are nodes and $\mathbf{E} \subseteq \mathbf{V} \times \mathbf{V}$ are directed edges. We say that $v_i$ is a parent of $v_j$ if and only if $(v_i, v_j) \in \mathbf{E}$, and denote as $Pa_i$ the parents of $v_i$. In most cases, the true causal graph is unknown, and the task of extracting a causal graph is called *causal discovery*.

When all variables are observable, we say there is *causal sufficiency*. Otherwise, if hidden variables are present, we have *causal insufficiency*. In causally insufficient scenarios it might not be possible to establish the direction of causation, and we can use a Maximal Ancestral Graph (MAG) (Zhang, 2008) to account for those ambiguities by using undirected edges. Before defining a MAG, we first recall what is an *inducing path*. Given a subset $\mathbf{L} \subset \mathbf{V}$, we say that a path is inducing relative to $\mathbf{L}$ if every vertex not in $\mathbf{L}$ is a collider on the path (apart from its endpoints), and every collider is an ancestor of at least one of the endpoints.

**Definition 1** (Maximally Ancestral Graph (Zhang, 2008))**.** *A mixed graph is called a maximal ancestral graph (MAG) if: 1) the graph does not contain any directed or almost directed cycles (i.e. it is Ancestral), and 2) there is no inducing path between any two non-adjacent vertices.*

A MAG can be obtained by marginalizing hidden variables in a DAG (Verma & Pearl, 2013). The resulting MAG describes a group of causal models preserving ancestral and independence relations between observable variables.

An additional challenge is *causal identifiability*: When specific assumptions on the structural mechanisms are not fulfilled (Shimizu et al., 2011; Hoyer et al., 2008), multiple causal graphs become statistically undistinguishable, unless prior knowledge (Zheng et al., 2024) or interventional data (Brouillard et al., 2020; Li et al., 2023) is available. When only observational data is available, it is often not possible to identify a unique causal graph (Mooij et al., 2016). In that case, we are limited to a set of plausible causal models called the Markov Equivalence Class (MEC), which is an equivalence class containing all causal graphs sharing the same set of conditional independencies. Consequently, causal discovery on causally insufficient scenarios using observational data often results in an equivalence class of MAGs (Spirtes, 2001; Rohekar et al., 2022), which can be graphically represented with a Partially Ancestral Graph (PAG), that we define below.

**Definition 2** (Partially Ancestral Graph)**.** *Let $\mathcal{M}$ be an arbitrary MAG. We can represent the MEC associated to $\mathcal{M}$ with a (PAG) $\mathcal{P}$, which is a partial mixed graph such that: (1) $\mathcal{P}$ shares the same adjacencies as $\mathcal{M}$, (2) An edge has an arrowhead (▶) in $\mathcal{P}$ iff it is shared by all MAGs in the MEC, (3) An edge has a tail (—) in $\mathcal{P}$ iff it is shared by all MAGs in the MEC, and (4) An edge has a circle ∘ in $\mathcal{P}$ iff it is neither an arrowhead or a tail.*

Classic algorithms yielding a PAG in causally insufficient scenarios are the Fast-Causal-Inference (FCI) (Spirtes, 2001) and the Iterative Causal Discovery (ICD) (Rohekar et al., 2022) algorithms. Both algorithms are constraint-based methods which iteratively remove edges when a conditioning set is found. That is, we can remove $u \to v$ when a conditioning set $\mathbf{Z}$ making them independent exists ($\exists \mathbf{Z} : u \perp\!\!\!\perp v | \mathbf{Z}$ with $u, v$, and $\mathbf{Z}$ disjoint). Inconveniently, the number of possible conditioning sets grows exponentially with the number of variables, and many constraint-based methods present severe scaling limitations (Feigenbaum et al., 2023). A valid approach for tackling high-dimensional data is to split variables into groups to be analyzed separately (Wu et al., 2025). A mathematically rigorous way of grouping variables is the one of *Causal Partitionings*, defined in (Zhang et al., 2022) and reported below.

**Definition 3** (Causal Partitioning)**.** *Let $\mathcal{G}(\mathbf{V}, \mathbf{E})$ be a graph with variable set $\mathbf{V}$ and edges $\mathbf{E} \subseteq \mathbf{V} \times \mathbf{V}$. A partitioning $\mathbf{P} = \{\mathbf{P}_1, \ldots, \mathbf{P}_M\}$ of $\mathbf{V}$ is a causal partitioning if and only if the following holds:*

1. *$\bigcup_{i=1}^{M} \mathbf{P}_i = \mathbf{V}$,*

2. *$\forall u, v \in \mathbf{V}$, if $u$ and $v$ are in separate partitions, then $u$ and $v$ are nonadjacent,*

3. *$\forall u, v \in \mathbf{V}$ that are nonadjacent in $\mathcal{G}$, they can be either:*

   - *Contained on separate partitions, i.e., $u \in \mathbf{P}_i$ and $v \in \mathbf{P}_j$ with $i \neq j$,*
   - *Contained on the same partition $\mathbf{P}_k$ which also contains their d-separating set $\mathbf{Z}$, such that $u \perp\!\!\!\perp v | \mathbf{Z}$ with $\mathbf{Z} \subset \mathbf{P}_k$ and $u, v$, and $\mathbf{Z}$ are disjoint.*

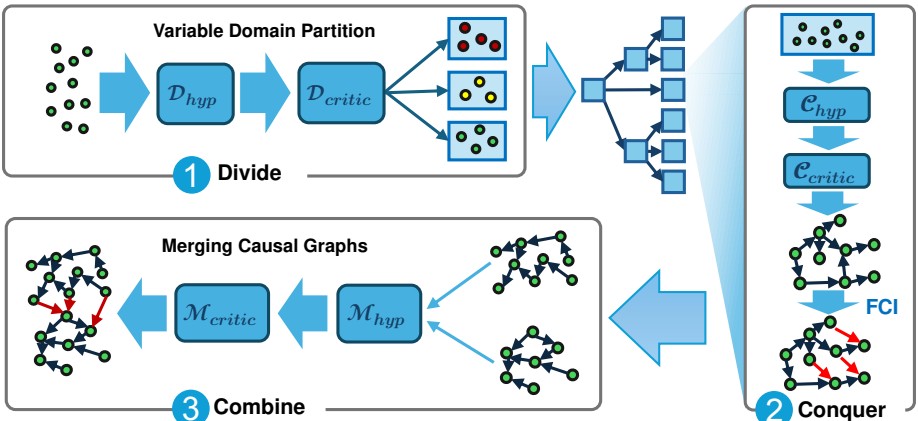

Figure 2: **Phases of the CAST Approach**. **1) Divide phase:** Causal variables (circles) are iteratively splitted into partitions (rectangles) that are likely causally connected. **2) Conquer phase:** A local causal graph is derived for each partition. **3) Combine phase:** Local causal graphs are merged.

The intuition behind causal partitionings is that variables are grouped in partitions reflecting how they are causally connected: If two variables are within the same partition, then they are causal neighbors. Importantly, a partition in a Causal partitioning can be further causally partitioned. Nested causal partitions can be arranged in a (causal) partition tree. In the ideal case, a causal partitioning can be computed exactly (Zhang et al., 2022). In practice, computing a causal partitioning is intractable, which forces heuristic estimates as in Zhang et al. (2022).

**Divide-and-Conquer:** The Divide-and-Conquer paradigm (Cormen et al., 2009) relies on recursively splitting problems into smaller sub-problems, and successively merging their solutions. Algorithms in this class are typically executed in 3 separate phases: 1) *Divide* phase, during which the problem is iteratively partitioned in sub-problems until they are easy enough to be solved, 2) a *Conquer* step during which each sub-problem is individually solved, 3) a *Combine* step where the solutions of the sub-problems are merged following the order of sub-division in reverse. The power of this class of algorithms is that, upon satisfying a set of mathematical properties, we have guarantees on the accuracy of the solution, and sometimes even on its optimality. Recently, Divide-and-Conquer strategies have been applied to prompting (Zhang et al., 2024).

RELATED WORK

**LLMs for Causality:** Formal causal reasoning of LLMs has been investigated in several studies (Willig et al., 2022; Kiciman et al., 2024; Jin et al., 2023; Zečević et al., 2023). While generally being able to perform simple causal reasoning tasks, studies generally find shortcomings, such as memorization and lack of generalization, in the causal reasoning capabilities of LLM. Further, Vashishtha et al. (2024) and Antonucci et al. (2023) explore to which extent LLMs can be taught causal reasoning, and show their capacity of extracting causal information from text. For practical applications, LLMs have been applied for causal discovery and in the design of randomized-control trials (Jiralerspong et al., 2024; Abdulaal et al., 2024; Tu et al., 2023; Kiciman et al., 2024; Le et al., 2025; Takayama et al., 2025). However, challenges persist in high-dimensional settings due to finite context windows and long edge-lists representing causal relationships, which motivates our divide-and-conquer approach.

**High-dimensional causality:** Real world data can exhibit high dimensionality and non-identifiable causal relationships, making most causal models computationally prohibitive and inaccurate (Tagliapietra et al.,

2025). Consequently, researchers explored gradient-based methods (Zheng et al., 2018; Sanchez et al., 2023), cluster-DAGs (Niu et al., 2022), divide-and-conquer approaches (Wu et al., 2025; Dong et al., 2025), and new variants of classic methods (Nazaret & Blei, 2024; Andrews et al., 2023). Further, Human-in-the-Loop systems are studied in Wang et al. (2025), da Silva et al. (2024) and Kitson & Constantinou (2023).

## 3 CAUSALSTEWARD: A MULTI-AGENTIC CAUSAL COPILOT FOR CAUSAL DISCOVERY

In this section we describe the architecture and core principles of CAST, which are further highlighted in Fig.1. First, we present the core components of CAST. Next, we dive into our Explain-Divide-Conquer-Combine approach. Lastly, we describe how CAST retrieves causal information.

**Human-in-the-Loop Causal Discovery:** CAST is a framework for interactive causal discovery. It assists the experts in building a causal graph through a Human-in-the-Loop system, while automating the retrieval of prior knowledge from external sources. Our approach enables a more natural interaction, overcoming the limitations of manual methods. For example, instead of tediously enumerating edge constraints, which is unfeasible with high-dimensional data, humans can specify patterns of causal relationship or provide only partial causal information. Moreover, CAST is responsible for translating human-provided information into the language of causality, and also for complementing the human when it is uncertain.

**Causal Discovery through Divide-and-Conquer:** One of the main challenges in causal discovery is dealing with high-dimensional data (Brouillard et al., 2024). CAST tackles high-dimensional causality with a divide-and-conquer approach, where variables are recursively assigned to smaller and more manageable partitions. To do so, CAST employs LLM agents to decompose the variables into separate smaller and causally-coherent clusters of variables. The clustering of variables is dynamic and iterative: clusters can be subdivided multiple times until the right level of granularity is reached. In 3.2 we define the criterion that allows CAST to adapt to the complexity of the task. Following, each sub-problem consists of estimating a local causal graph for each partition of variables. The core intuition is that LLMs perform better on smaller and more targeted tasks (Khot et al., 2023). In practice, variables within each group should be single-themed and refer to a single causal phenomenon. This facilitates the agents during the conquer step, as they can reflect more in detail and perform more targeted RAG (or human) queries to retrieve the necessary causal information. As we will show in the Sec.5, variable partitioning makes the individual sub-tasks more intuitive for the agents. In practice, CAST is executed in four phases: *Explain*, *Divide*, *Conquer*, and *Combine*. These phases are carried out by a set of specialized agents, each one responsible for a specific task in the causal discovery process. First, during the **(1) Explain phase**, CAST gathers all necessary information before starting any discovery process, then the **(2) Divide phase** repeatedly partitions the variable set **V**. Afterwards, the **(3) Conquer phase** produces a local causal graph for every partition. Finally, the **(4) Combine phase** merges all the local causal graphs. In Fig.2 we provide a graphic depiction of CAST. We highlight that all our agents can be provided with RAG tools and human interaction to dynamically retrieve all the necessary causal information.

### 3.1 EXPLAIN PHASE

Prior to the divide-conquer-combine phases, CAST implements an explanation step where an Explainer agent $\mathcal{E}$ gathers information about the task and complements the information provided by the user. By doing so, it decreases the ambiguity about the meaning of each variable. First, the available meta-data (variable labels and dataset description) is provided to the agent. Next, $\mathcal{E}$ outputs an expanded description with details on every single variable. We denote this new information as $I$ and write this step as

$$I = \mathcal{E}(\mathrm{X}_{labels}, \mathbf{p}_{explain}), \tag{1}$$

where $X_{labels}$ are the variable labels, and $\mathbf{p}_{explain}$ is a prompt containing the directions for the agent plus a task description provided by the user.

## 3.2 DIVIDE PHASE

The Divide phase recursively partitions the variable set $\mathbf{V}$ with the aim of grouping causally related and semantically similar variables together. This step is carried out by two agents: the Divide-hypothesis agent $\mathcal{D}_{hyp}$ and the Divide-critic agent $\mathcal{D}_{critic}$. Both agents leverage the information gathered by the explainer agent to estimate a causal partitioning of $\mathbf{V}$. Mathematically, given a partition $\mathbf{P}_i \subseteq \mathbf{V}$ and a divide prompt $\mathbf{p}_{div.hyp.}$, the Divide-hypothesis agent $\mathcal{D}_{hyp}$ first proposes a possible partitioning,

$$\{\mathbf{P}_{i,1}, \ldots, \mathbf{P}_{i,N}\} = \mathcal{D}_{hyp}(\mathbf{P}_i, \mathbf{p}_{div.hyp.}), \tag{2}$$

where $\mathbf{P}_{i,k}$ denotes the k-th partition of $\mathbf{P}_i$, such that $\bigcup_k \mathbf{P}_{i,k} = \mathbf{P}_i$. Different partitions can overlap, i.e. $\mathbf{P}_{i,k} \cap \mathbf{P}_{i,l}$ might be non-empty for some $k$ and $l$. Following, a second Divide-critic agent is tasked to review (using the prompt $\mathbf{p}_{div.critic}$) the partitioning and apply corrections if necessary,

$$\{\tilde{\mathbf{P}}_{i,1}, \ldots, \tilde{\mathbf{P}}_{i,N}\} = \mathcal{D}_{critic}(\{\mathbf{P}_{i,1}, \ldots, \mathbf{P}_{i,K}\}, \mathbf{p}_{div.critic}). \tag{3}$$

The causal partitioning process continues recursively until $\mathcal{D}_{hyp}$ stops partitioning further and concludes that the right level of granularity for causal discovery has been reached. In detail, $\mathcal{D}_{hyp}$ and $\mathcal{D}_{critic}$ are queried again on every partition with more than a number $k$ of variables, and they have to decide whether to partition further or not. This iterative causal partitioning of the general task in simpler sub-tasks can be arranged into a partition tree, where each node describes a partition and its children are sub-partitions (Fig 2).

## 3.3 CONQUER PHASE

The conquer phase performs LLM-aided causal discovery. Once the divide phase estimates a causal partitioning, a conquer step derives a local causal graph for the variables within each partition that is a leaf in the causal partition tree. First, CAST employs two LLM agents to formulate a causal hypothesis in form of a graph. Next, a data-driven discovery step utilizes the causal hypothesis as a constraint to improve causal discovery.

**1) Hypothesis Generation:** After being provided with a description of the dataset, a hypothesis agent $\mathcal{H}$ formulates an initial causal hypothesis $\mathcal{C}_{hyp}$ for the variables within the partition. Given a partition $\mathbf{P}_i$, we have

$$\mathcal{G}_i^{hyp} = \mathcal{C}_{hyp}(\mathbf{P}_i, \mathbf{p}_{hyp}). \tag{4}$$

**2) Critic Evaluation:** A critic agent $\mathcal{C}$ evaluates the causal hypothesis $\mathcal{G}^{hyp}$ and refines it based on its internal prior and available external information. Additionally, the agent is prompted to focus on being right at the interventional and at the counterfactual level.

$$\mathcal{G}_i^{critic} = \mathcal{C}_{critic}(\mathcal{G}_i^{hyp}, \mathbf{p}_{critic}) \tag{5}$$

**3) Data-Driven Discovery:** A causal discovery algorithm is used to insert additional edges to $\mathcal{G}_i^{critic}$. In practice, we use the edges in $\mathcal{G}_i^{critic}$ to bootstrap a constraint-based causal discovery algorithm $\mathcal{F}$. Mathematically,

$$\tilde{\mathcal{G}}_i^{critic} = \mathcal{F}(\mathtt{D}_i, \mathcal{G}_i^{critic}), \tag{6}$$

where $\mathtt{D}_i$ is a subset of the input dataset corresponding to the variables in $\mathbf{P}_i$, and $\tilde{\mathcal{G}}_i^{critic}$ is the causal graph discovered by $\mathcal{F}$. This discovery process complements the prior information provided by LLMs. In real scenarios, causal relationships can be non-identifiable, and prior constraints improve identifiability and overall performance (Zheng et al., 2024). In turn, the discovery process may find valid relationships that the LLM failed to extract from knowledge sources.

### 3.4 COMBINE PHASE

The Combine phase merges the local causal graphs generated during the Conquer phase into a single global causal graph. Since edges within a partition are already estimated, the combine phase is responsible for merging those graphs. As written in Sec.4, if the Divide phase yielded a correct causal partitioning, performing the union of all local causal graphs is sufficient to obtain a consistent global causal graph. However, a practical challenge persists: if the causal partitioning operation during the divide phase is not accurate, the local causal graphs might not overlap, leading to disconnected components. Therefore, when merging two or more subgraphs, it is important to assess for edges between nodes in one group and another. For this phase, we adopt a procedure similar to the conquer step. This time, a merge-hypothesis agent $\mathcal{M}_{hyp}$ is given a description of the groups, and is queried to generate a causal hypothesis for the missing edges that connect the local causal graphs, which we call *merging causal hypothesis*. Similarly to before, once a first merging causal hypothesis is obtained, a merge-critic agent $\mathcal{M}_{critic}$ has the task to refine it. Given M local subgraphs, we have

$$
\begin{aligned}
\mathcal{G}_{comb}^{hyp} &= \mathcal{M}_{hyp}(\mathcal{G}_1, \ldots, \mathcal{G}_M, \mathbf{p}_{comb}), \\
\mathcal{G}_{comb}^{critic} &= \mathcal{M}_{critic}(\mathcal{G}_{comb}^{hyp}, \mathbf{p}_{comb}),
\end{aligned}
\tag{7}
$$

where $\mathcal{G}_{comb}^{hyp}$ and $\mathcal{G}_{comb}^{critic}$ are graphs with nodes $\mathbf{V}_{comb} = \bigcup_{i=1}^{M} \mathbf{V}_i$. The combine operation is iterated by following the order of subdivision in reverse. In other words, we follow the structure of the partition tree from its leaves to the root.

### 3.5 RETRIEVAL OF EXTERNAL KNOWLEDGE

An ideal human-in-the-loop copilot can efficiently acquire information by balancing RAG for general knowledge and targeted queries to the human for context-specific information. Therefore, every agent in CAST can gather external knowledge either via retrieval-augmented generation (RAG) or via human interaction. For the **Human-in-the-loop**, all the agents have the capacity to query a human for additional information. The interaction is made on-demand, and takes the form of a tool-call. Similarly, we also provide to our agents the capacity to dynamically summon an **Agentic RAG** system on demand. Our RAG system can perform a web search and retrieve relevant information from the internet. We describe the system in detail in appendix C and present an exemplary application of CAST on the ASIA dataset (Lauritzen & Spiegelhalter, 2018) in Fig. 3.

## 4 THEORETICAL ANALYSIS

We first study the ideal case. Next we analyze a general scenario. Throughout our analysis, we will always assume *Causal Faithfulness*. Assuming that $\mathcal{D}_{hyp}$ and $\mathcal{D}_{critic}$ derive a partitioning that is causal by Def.3, we can prove that CAST can reach asymptotically the correct MEC.

**Theorem 1** (Consistency for an ideal causal partitioning)**.** *Let $\mathcal{G}_i$ be the local PAG obtained after performing the conquer phase on variables in partition $\mathbf{P}_i$, and denote with $\mathcal{G}$ the graph obtained by performing the union of all local causal graphs $\mathcal{G}_i$, $\forall i$. If all conquer phase is consistent, then also $\mathcal{G}$ is a PAG representing the MEC entailed by the observational data.*

In our case, since FCI is consistent in the large-sample limit, Th.1 implies that CAST inherits said asymptotic consistency. In general applications, however, Divide agents might not yield a correct causal partitioning and the conquer agents might have inaccuracies. Therefore, we provide a general analysis on how the total SHD is influenced by each component in non-ideal scenarios.

**Theorem 2.** *Let $\mathbf{P}$ be a partitioning induced by $\mathcal{D}_{hyp}$ and $\mathcal{D}_{critic}$. Define $SA$ as the number of spurious adjacencies, $E_{cut}$ and $N_{NotIntra}$ the number of inter-partition edges present resp. absent in the ground*

| Dataset | | Small | | | Medium | | |
| --- | --- | --- | --- | --- | --- | --- | --- |
| | Method | F1 ↑ | Prec.↑ | SHD ↓ | F1 ↑ | Prec.↑ | SHD ↓ |
| | FCI | 0.028(0.012) | 0.021(0.005) | 221.4(3.36) | | Intractable | |
| | XGES | 0.056(0.009) | 0.037(0.006) | 448.8(14.89) | 0.014(0) | 0.211(0) | 547(0) |
| | GranDAG | 0.002(0.003) | 0.018(0.040) | 113.4(2.51) | 0(0) | 0(0) | **544.4(5.77)** |
| | CausalCopilot | 0.078(0.012) | 0.260(0.100) | 117.4(6.4) | 0.014(0) | 0.211(0) | 547(0) |
| | BOSS | 0.054(0.004) | 0.029(0.002) | 864.20(18.60) | 0.057(0.005) | 0.030(0.002) | 1,817.4(48.3) |
| | LLM-BFS(o3-mini) | 0.283(0.000) | **1.000(0.000)** | 315.2(0.0) | — | — | — |
| | Pairwise(Qwen3-14B) | 0.153(0.008) | 0.093(0.005) | 583.50(11.1) | — | — | — |
| o3-mini | CS (HITL+RAG) | 0.303(0.076) | 0.454(0.123) | 118.4(11.35) | 0.141(0.064) | 0.357(0.119) | 579.6(25.01) |
| | CS (HITL) | 0.304(0.062) | 0.485(0.101) | 112.2(10.40) | 0.175(0.084) | 0.499(0.127) | 558.8(15.498) |
| | CS (RAG) | 0.235(0.093) | 0.373(0.118) | 121.4(6.34) | 0.209(0.089) | 0.518(0.201) | 571.6(45.845) |
| o4-mini | CAST (HITL+RAG) | 0.142(0.027) | 0.217(0.063) | 140.4(13.00) | 0.086(0.042) | 0.271(0.113) | 568.2(20.71) |
| | CAST (HITL) | 0.139(0.056) | 0.326(0.149) | 126.8(15.93) | 0.103(0.039) | 0.392(0.083) | 547.0(6.89) |
| | CAST (RAG) | 0.113(0.039) | 0.169(0.047) | 145.0(14.35) | 0.113(0.049) | 0.296(0.094) | 594.2(81.69) |
| Qw.3 14B | CAST (HITL+RAG) | 0.424(0.082) | 0.724(0.169) | 103.00(10.07) | **0.274(0.052)** | **0.656(0.110)** | 558.0(21.86) |
| | CAST (HITL) | **0.483(0.062)** | 0.608(0.069) | **102.4(17.27)** | 0.266(0.019) | 0.508(0.222) | 610.80(68.00) |
| | CAST (RAG) | 0.426(0.033) | 0.656(0.176) | 111.4(14.44) | 0.211(0.106) | 0.485(0.129) | 568.4(23.64) |

Table 1: **Results on CausalMan Small and Medium.** CAST variants largely outperform other baselines on Small and strong competitive results are achieved on Medium. CausalCopilot executed FCI on Small and XGES on Medium. FCI, LLM-BFS and LLM-Pairwise could not run on Medium.

truth. Let $FPR_{\mathcal{C}}$ denote the false-positive rate of Conquer agents, and let $Rec_{\mathcal{M}}$ and $FDR_{\mathcal{M}}$ denote respectively the recall and false-discovery rate of Merge agents. Then, as $n \to \infty$, the expected total structural Hamming distance satisfies

$$\mathbb{E}[SHD_{total}] \approx (SA + N_{NotIntra} \cdot FPR_{\mathcal{C}}) + (E_{cut}(1 - Rec_{\mathcal{M}}) + m \cdot FDR_{\mathcal{M}}). \tag{8}$$

We observe above how the performance of CAST is linked to the accuracy of $\mathcal{D}_{hyp}\&\mathcal{D}_{critic}$ to derive a correct causal partitioning. Still, merge agents $\mathcal{M}_{hyp}\&\mathcal{M}_{critic}$ provide a safety-net that recover edges lost during the partitioning operation. Further, the performance of CAST is tied to the capacity of LLMs to provide the correct edge constraints to the causal discovery algorithm $\mathcal{F}$ during the conquer phase. Therefore, it is crucial for CAST to access clear and reliable sources of information, which is why our agents leverage RAG and HITL, as described in Sec.3.5. We remark that the human-in-the-loop influences all agents performance. All our proofs are in appendix A.

## 5 EXPERIMENTS

We benchmark on the CausalMan dataset (Tagliapietra et al., 2025) and on the Neuropathic-Pain dataset (Tu et al., 2019). We evaluate $F_1$, precision, and Structural Hamming Distance (SHD). We repeat experiments with 5 different seeds, and compute mean and standard deviation for every metric. We also count the number of tool calls to RAG and HITL. HITL experiments on CausalMan have been carried on with the support of real domain experts in manufacturing. Extended tables in E.

**Large-Language Models:** We test three different LLMs: GPT 4o-mini (OpenAI, 2024a), o3-mini (OpenAI, 2024b), and Qwen3-14B (Qwen, 2025). Prompt templates are in appendix G. Graphs are encoded in form of node- and edge-lists (Fatemi et al., 2023). No fine-tuning is performed.

**Ablations:** We analyze the contribution of human interaction by testing three variants of CAST. One with both HITL and RAG active (HITL+RAG), and two with only either HITL or RAG enabled. CAST(RAG)

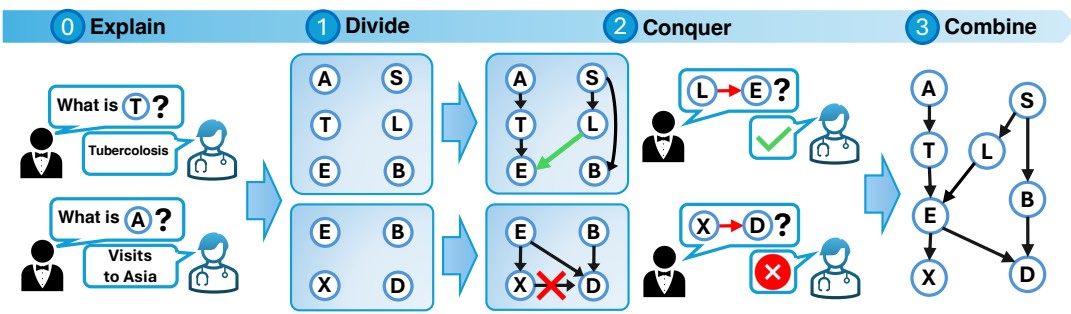

Figure 3: Illustration for CAST on the ASIA dataset (Lauritzen & Spiegelhalter, 2018). First, the **Divide phase** partitions the variable set **V** in $\mathbf{P}_1 = \{A, T, E, L, S, B\}$ and $\mathbf{P}_2 = \{E, X, D, B\}$. Next, during the **Conquer phase**, a local causal graph is estimated. Further, agents query the human to check whether $L \to E$, and if $X \to D$. After confirming the first and denying the latter, a causal discovery algorithm finalizes a local causal graph $\mathcal{G}_1$ for $\mathbf{P}_1$ and $\mathcal{G}_2$ for $\mathbf{P}_2$. Next, the **Combine phase** merges $\mathcal{G}_1$ and $\mathcal{G}_2$ into the global graph $\mathcal{G}$.

features no human interaction, and is thus completely automatic. We additionally ablate with respect to the Divide&Conquer approach, with respect to the Critic mechanism. In appendix B we also include an ablation with respect to the partitioning hyperparameter $k$.

**Baselines:** We compare with XGES (Nazaret & Blei, 2024), GranDAG (Lachapelle et al., 2020), (Andrews et al., 2023), FCI (Spirtes, 2001), BOSS (Andrews et al., 2023). We include also LLM-based approaches as CausalCopilot (Wang et al., 2025), LLM-BFS (Jiralerspong et al., 2024), and LLM-Pairwise (Kiciman et al., 2024).

## 6 EMPIRICAL ANALYSIS

We focus our analysis on the performance, scalability and capacity to use prior knowledge from RAG and HITL. Therefore, we assess CAST with the following research questions: **(Q1)** Can CAST tackle causal discovery?; and **(Q2)** Can CAST scale?; and **(Q3)** Do computational requirements scale tractably with the size of the graph?; and finally **(Q4)** How beneficial are RAG and HITL to CAST?

### 6.1 RESULTS ON MANUFACTURING AND NEUROPATHIC PAIN DATA

**Setup:** We test on 2 variants of the CausalMan benchmark (Tagliapietra et al., 2025): CausalMan Small (53 nodes, 108 edges), and CausalMan Medium (186 nodes, 553 edges), which exhibit causal insufficiency, mixed data-types, and diversified causal mechanisms. We also test on the Neuropathic-Pain dataset (222 nodes, 770 edges), which is causally sufficient with binary variables. All those benchmarks have an expressive variable naming where valuable information can be extracted with LLMs. Further, they have repeated causal sub-structures. More details in appendix F.

**Results:** See Table 1 (extended in appendix B). CAST considerably improves over purely data-driven baselines, outperforming for precision and F1. Improvements for SHD are not as significant, and GranDAG is close. Overall, results answer **(Q1)** positively, with margin for improvement in terms of recall, which signals that CAST tends to discover sparse graphs. Regarding scalability, the $F_1$ decrease in performance from CausalMan Small to Medium is only sub-linear (by a factor of $\approx 2$) for CAST, whereas most data-driven baselines (XGES, GranDAG, CausalCopilot) degrade by several times. BOSS exhibits stronger ro-

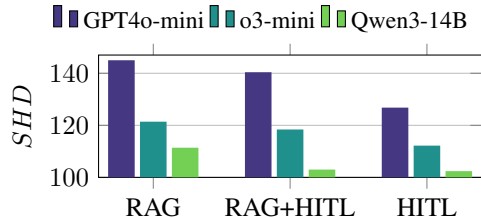

| | Method | $F_1 \uparrow$ | Prec.$\uparrow$ | SHD $\downarrow$ |
|---|---|---|---|---|
| Small | CAST | **0.42(0.08)** | **0.72(0.17)** | **103.0(10.1)** |
| | no D&C/FCI | 0.34(0.06) | 0.34(0.08) | 199.0(20.3) |
| | no Critic | 0.29(0.09) | 0.38(0.16) | 139.6(20.9) |
| | no Explainer | 0.239(0.107) | 0.45(0.12) | 120.20(16.94) |
| Medium | CAST | 0.27(0.05) | **0.66(0.11)** | 558.0(21.9) |
| | no D&C/FCI | — | — | — |
| | No Critic | **0.31(0.02)** | 0.56(0.13) | 588.0(43.2) |
| | No Explainer | 0.15(0.04) | 0.59(0.16) | **540.00(19.89)** |

Figure 4: **1) SHD on CausalMan Small for different LLMs (left).** Models good at instruction-following (Qwen3 or GPT4o-mini) can use RAG and HITL to increase their performance. **2) Ablations of D&C and Critic agents (right).** Removing D&C prevents scaling: without partitioning, inference slows and accuracy drops. Without Critic agents, precision and SHD degrade because wrong causal edges remain uncorrected. Results on Qwen3-14B.

bustness, but absolute metrics are still inferior with respect to CAST. Overall, results indicate that CAST scales robustly while keeping good performance, affirming **(Q2)**. We also attribute this to the redundancy in CausalMan Medium: It contains many repeated patterns in variable naming/metadata, which are easy for LLMs to detect. The (HITL+RAG) variant provides a good trade-off between automation and control, which becomes more evident when using models with good instruction-following capabilities. Although they are prompted to make use of tools and balance HITL with RAG, we see how o3-mini struggles to follow those directions. For Qwen3-14B, instead, we see a significant improvement when the Human is inserted into the system, supporting **(Q4)** if the right LLMs are used. We notice that the closest baselines are the LLM-based. For LLM-BFS we observe a perfect precision, which is counterbalanced by low recall and high SHD, signaling a very conservative approach in inserting causal edges. On the computational side, most classic methods (e.g. FCI) are intractable in high-dimensional settings. Our divide-and-conquer approach maintains a sub-quadratic growth from CausalMan Small to Medium both in token usage and runtime. Runtime indeed stays under 2 hours for all LLMs (Sec.B.3). Therefore, we confirm **(Q3)**. LLM-based baselines are limited in large-scale settings: LLM-pairwise requires $\mathcal{O}(D^2)$ queries (with $D$ = number of variables), which makes it applicable on CausalMan Small, but unfeasible on other benchmarks. LLM-BFS has similar scaling behavior, and is also stopped by the limited context window.

**Ablation for Divide&Conquer** In Fig.4, we ablate with respect to the divide&Conquer mechanism. By disabling the Divide phase, we encounter two major limitations: 1) The agents $\mathcal{C}_{hyp}$ and $\mathcal{C}_{critic}$ must now estimate a causal graph for the whole variable set $\mathbf{V}$, which leads to context overflows and slower inference, and 2) FCI is also required to estimate a causal graph for $\mathbf{V}$ as well, which is computationally prohibitive. To carry on the ablation, therefore, we deactivated also FCI, resulting in a Conquer phase solely based on LLMs. Still, without Divide&Conquer, experiments on CausalMan Medium and Neuropathic were not feasible, similarly to LLM-BFS and LLM-Pairwise. On CausalMan Small, the Divide&Conquer approach is still beneficial for SHD and precision.

**Ablation for Critic Agents** By removing $\mathcal{D}_{critic}$, $\mathcal{C}_{critic}$, and $\mathcal{M}_{critic}$, we see in Table 4 that precision and SHD drop consistently on all datasets. Hypothesis agents $\mathcal{D}_{hyp}$, $\mathcal{C}_{hyp}$, or $\mathcal{M}_{hyp}$ alone tend to generate denser graphs with higher recall (coverage), but lower precision. We deduce that critic agents enable the control of false positives (better $FPR_{\mathcal{C}}$), acting as a filter that improves factuality.

**Ablation for Causal Discovery Algorithm** During the Conquer Phase, $\mathcal{C}_{critic}$ outputs a set of edges that are used as constraints to a causal discovery algorithm. To validate our approach, we ablated with respect to the Causal Discovery algorithm, and now we test also the gradient-based DAGMA (Bello et al., 2022). We additionally test both DAGMA and FCI with and without edge constraints. For DAGMA, those constraints are

| Method | $F_1 \uparrow$ | Prec.$\uparrow$ | SHD $\downarrow$ |
|---|---|---|---|
| DAGMA | 0.1(0.3) | 0.1(0.3) | 4.5(1.8) |
| DAGMA+Agents | **0.6(0.3)** | **0.4(0.4)** | **3.1(2.9)** |
| FCI | 0 | 0 | 4.6(4.7) |
| FCI+Agents | 0.3(0.4) | 0.4(0.5) | 4.3(5.3) |

| Method | $F_1 \uparrow$ | Prec.$\uparrow$ | SHD $\downarrow$ |
|---|---|---|---|
| FCI | 0.059(0.011) | 0.054(0.009) | 84.4(3.6) |
| GranDAG | 0.032(1e-4) | 0.233(0.037) | **59.4(0.9)** |
| XGES | 0.053(1e-4) | 0.091(1e-4) | 81(0.0) |
| BOSS | 0.091(1e-4) | 0.081(1e-4) | 82(0.0) |
| CAST | **0.411(0.086)** | **0.315(0.059)** | 96.6(11.81) |

Figure 5: **1)** Ablation with respect to tha Causal Discovery algorithm used during the Conquer Phase. "+Agents" indicates that the edge constraints obtained by $\mathcal{C}_{hyp.}$ and $\mathcal{C}_{critic}$ are used. Using the agents constraints improved performance both for FCI and DAGMA. **2)** Results on CausalChambers real-world dataset.

inserted post-training. Metrics are evaluated for each individual partition, and averaged across all partitions and across all runs. Results in Table 5 show that both algorithms improve upon using the agents' constraints.

## 6.2 REAL-WORLD CASE-STUDIES

Finally, we apply CAST (with GPT-5) to the CausalChambers benchmarks (Gamella et al., 2024), and focus on the light-tunnel device. Results are in 5. We obtained a True-Positive-Ratio of 0.89 and a False-Positive-Ratio of 0.04, showing that CAST can valuably support experts during causal discovery. During execution, it performed multiple HITL queries, focused (at explain-phase) on clarifying some cryptic naming of variables, and later (during conquer-phase) on gathering details on the physical system.

Interestingly, the resulting graph this time is denser than the ground truth, and the conquer agents often provide coherent arguments for those additional links. This leads to higher SHD (76), which occurs because the model proposes additional causal sub-structures related to the electronics of the device, whose are absent in the ground truth graph. Those edges might either be unmodeled effects of the system, or overly detailed hypotheses (i.e. negligible causal relationships), which in both cases highlight CAST's capacity to retrieve potential relevant domain mechanisms that expert can then evaluate afterwards. Additional case study on the Earthquake dataset is reported in Appendix B.

## 7 CONCLUSIONS

CAST is an AI copilot and to our knowledge it is the first one combining a Human-In-The-Loop approach, a multi-agent architecture, and the Divide-and-Conquer paradigm for causal discovery. In our experiments we demonstrated considerable performance improvements over existing data-driven methods and empirically showed how CAST can integrate human feedback into causal discovery.

**Limitations and future Work:** The performance of CAST is limited by the availability and quality of prior information. Additionally, CAST could integrate retrieval-augmented generation (RAG) from more varied sources of prior knowledge, which could further enhance its capabilities. Indeed, CAST can be extended to use a wider set of tools and different LLM agents. Finally, this work could be extended beyond causal graphs, aiming towards more general causal reasoning tasks.

## ETHICS STATEMENT

We acknowledge that the behavior of CAST , being LLM-based, is sensible to data-leaks and/or bias in the content retrieved from the internet. Further, human accountability is necessary for who uses such systems, who need to be aware of the privacy risks of providing sensible data to agents.

**LLM-Usage for writing**   We acknowledge that LLM-based tools were used to improve the writing of this manuscript. The usage was limited to grammatical suggestions, phrase structure, and word choice. The design of the method or the experimental setting, instead, were not aided by any LLM.

## REPRODUCIBILITY STATEMENT

We provide the complete code-base to run each experiment that is present in this manuscript, and we report in Appendix E the computational resources that were used including hardware, runtime and token-count. For HITL experiments, attempts to reproduce results should take into account that we conducted such experiments with the help of human domain experts. This is a necessary requirement to be as close as possible to how CAST is used in new real world applications.

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

APPENDIX FOR "CAUSALSTEWARD: AN AGENTIC DIVIDE-CONQUER-COMBINE COPILOT FOR CAUSAL DISCOVERY"

## A  THEORY

We first provide a theory describing the ideal case, i.e. what happens when we have perfect agents which can derive a causal partitioning satisfying Def.3 without any error, while also having a perfect conquer phase. Next, we provide a generalization, where we provide error bounds and also an expression for the final SHD in terms of the performance of each agent.

### A.1  IDEAL CASE OF PERFECT CAUSAL PARTITIONING

We start from the assumption that the conquer phase is consistent in the infinite data limit.

**Assumption 1** (Consistency of the Conquer phase). *In the infinite data limit, for $n \to +\infty$, the conquer phase yields a PAG representing the local MEC $M_i$ for the variables within $\boldsymbol{P}_i$.*

Assumption 1 is valid for algorithms such as FCI, and is still valid if edge constraints provided by the LLMs are correct. When consistency of the conquer phase is assumed, then consistency of the overall method follows, as we prove in the next theorem.

**Theorem 3** (Consistency of the Combine phase). *Let $\mathcal{G}_i$ be the local causal graph (a PAG) obtained after performing the conquer phase on variables in partition $\boldsymbol{P}_i$, and denote with $G$ the graph obtained by performing the union of all local causal graphs $\mathcal{G}_i$, $\forall i$. Under assumption 1, $\mathcal{G}$ is a PAG representing the MEC entailed by the observational data.*

*Proof.* First, we study how the MEC of two partitions $\boldsymbol{P}_1$ and $\boldsymbol{P}_2$ can be decomposed. Next, we need to show that a causal partitioning is sufficient, i.e. it includes all those c.i. statement in the MEC. Extensions to multiple partitions automatically follows by studying all possible pairs.

First, we show that the set of c.i. statements of $\boldsymbol{P}_1 \cup \boldsymbol{P}_2$ can be partitioned into 3 (possibly overlapping) sets:

- $\mathbf{C}_1$ : All c.i. statements within $\boldsymbol{P}_1$, such as $u \perp v \mid \mathbf{Z}$, $\forall u, v, \mathbf{Z} \in \boldsymbol{P}_1$.

- $\mathbf{C}_2$ : All c.i. statements within $\boldsymbol{P}_2$, such as $u \perp v \mid \mathbf{Z}$, $\forall u, v, \mathbf{Z} \in \boldsymbol{P}_2$.

- $\mathbf{C}_{1-2}$ : All c.i. statements between variables in separate partitions, such as $u \perp v \mid \mathbf{Z}$, $u \in \boldsymbol{P}_1, v \in \boldsymbol{P}_2$, and $u \notin \boldsymbol{P}_2, v \notin \boldsymbol{P}_1$, with $\mathbf{Z} \subset \boldsymbol{P}_1 \cup \boldsymbol{P}_2$.

In other words, we can rewrite the set of conditional independencies as $\mathbf{C} = \mathbf{C}_1 \cup \mathbf{C}_2 \cup \mathbf{C}_{1-2}$. To complete the proof, we need to show that all those sets follow from a Causal Partitioning.

For all valid causal partitionings obeying def.3, all those C.I. statements described in (1), (2), and (3) are conserved. Indeed, for $\mathbf{C}_1$ and $\mathbf{C}_2$, it follows from def.3, point (3b): $\forall u, v \in \boldsymbol{P}_i$ then also $\mathbf{Z} \subset \boldsymbol{P}_1$. For $\mathbf{C}_{1-2}$, those c.i. statements are "encoded" in def.3. In point (3a), two nodes $u$ and $v$ in separate partitions are not connected $\to u$ and $v$ are conditionally (or marginally) independent, i.e. $\exists \mathbf{Z} \subset \boldsymbol{P}_1 \cup \boldsymbol{P}_2$.

Since the sets $\mathbf{C}_1$, $\mathbf{C}_2$, and $\mathbf{C}_{1-2}$ are all entailed by the definitions of causal partitioning, it follows that all c.i. statements of $P_1 \cup P_2$ are conserved under a causal partitioning. $\square$

**Can $\mathbf{C}_1$, $\mathbf{C}_2$, and $\mathbf{C}_{1-2}$ overlap?**   $\mathbf{C}_1$ and $\mathbf{C}_2$ can overlap in those cases where both variables and their conditioning set are contained in the intersection $\mathbf{P}_1 \cap \mathbf{P}_2$, i.e., $u \perp v | \mathbf{Z}$ with $u, v, \mathbf{Z} \in \mathbf{P}_1, \mathbf{P}_2$.

**Example 1.** *A partitioning for the graph $A \to B \to C$ into two partitions would be $\boldsymbol{P}_1 = \{A, B\}$ and $\boldsymbol{P}_2 = \{B, C\}$. The conditional independence $A \perp C | B$, as written above, follows from the partitioning itself i.e. from having $A$ and $C$ in separate partitions.*

## A.2   ERROR-BOUNDS FOR APPROXIMATIONS OF CAUSAL PARTITIONINGS

Given the Divide&Conquer approach, CAST can be analysed in 2 parts: Inter-Partition level, and Intra-Partition level. The former is associated with edges between nodes in separate partitions, and the latters with edges within a partition, that belong to the individual sub-graphs.

**Inter-Partition Analysis**   To analyze how errors are generated during the partitioning operation, we define the Causal-Cut as

$$E_{cut} = |\{(u, v) \in \mathbf{E} | \exists \mathbf{P}_i, \mathbf{P}_j \in \mathbf{P} \text{ with } i \neq j \text{ such that } u \in \mathbf{P}_i \text{ and } v \in \mathbf{P}_j\}| \tag{9}$$

where $E_{cut}$ is the number of causal edges "severed" by the partitioning operation, i.e. the size the cut. The causal cut measures the number of edges bridging partitions that are lost during the partitioning operation. We remark that $E_{cut}$ is computed between a partitioning and a causal graph, therefore it ignores eventual merging heuristics.

Still, our analysis must include the effect of Merge agents $\mathcal{M}_{hyp}$ and $\mathcal{M}_{critic}$, which are responsible for inserting additional edges between partitions. Those results in added edges which might increase the number of true positives, but also the number of false positives in case of bad guesses. We define with $H_{correct}$ the number of true positives (TP) guessed by the heuristic, and with $H_{incorrect}$ the sum of false negatives (FN) and false positives (FP) guessed by the merge agents. In the following, we assume $E_{cut}, H_{correct}, H_{incorrect}$ to be random variables that count such quantities.

Our first result characterizes the partitioning operation itself, which models the number of false positives in the learned graph.

**Theorem 4** (Partitioning-induced False Positives). *Let $\boldsymbol{P}$ be a partitioning generated by the divide agents $\mathcal{D}_{hyp}$ and $\mathcal{D}_{critic}$, and denote with $\mathcal{G}_{learned}$ and $\mathcal{G}^\star$ respectively the PAG discovered by CAST and the ground truth PAG. Further, let $\bar{p}$ be the average probability of $\mathcal{M}_{hyp}$ and $\mathcal{M}_{critic}$ discovering an inter-partition edge. Then, the inter-partition SHD in $\mathcal{G}_{learned}$ can be written as*

$$SHD_{inter} = E_{cut} - H_{correct} + FP_{inter} \tag{10}$$

*Further, if we assume that the bernoulli probabilities of guessing an inter-partition edge are equal and independent, it also holds in expectation that*

$$\mathbb{E}[SHD_{inter}] = \mathbb{E}[E_{cut}](1 - \bar{p}) + \mathbb{E}[FP_{inter}]. \tag{11}$$

*Proof.* First of all, we model the probability of guessing an inter-partition edge with $p_e$, and assume that each edge discovery is independent from the other. It follows that a random variable counting the discovered inter-partition edges can be modeled as a sum of independent bernoulli random variables. In other words, we have

$$\bar{p} = \frac{\sum_{e \in E_{cut}} p_e}{E_{cut}} = \mathbb{E}\left[\frac{H_{correct}}{E_{cut}}\right]. \tag{12}$$

Next, we study which kind of errors arise from the partitioning operation. Since there is no "intra-partition" discovery phase during which we learn causal edges, there is no possibility to insert False Negatives besides

the Merge Heuristic, which acts as a corrective mechanism. Therefore, we can quantify the number of false negatives with $E_{cut}$ (which are the false negatives derived from the partitioning), minus those edges recovered by the heuristic, which are $H_{correct}$. So, we have

$$FN_{inter} = E_{cut} - H_{correct} \tag{13}$$

By adding $FP_{inter}$ at both sides, it yields

$$FN_{inter} + FP_{inter} = E_{cut} - H_{correct} + FP_{inter} \tag{14}$$

$$SHD_{inter} = E_{cut} - H_{correct} + FP_{inter} \tag{15}$$

The expected value of $H_{correct}$ is the sum of the expectations of $E_{cut}$ individual bernoulli variables $p_e$. Assuming that $p_e = p \in \mathbb{R}$, we have $\mathbb{E}[H_{correct}] = \mathbb{E}[E_{cut}] \cdot p$. By substitution, and by linearity of expectations, we have

$$\mathbb{E}[SHD_{inter}] = \mathbb{E}[E_{cut} - H_{correct} + FP_{inter}], \tag{16}$$

$$\mathbb{E}[SHD_{inter}] = \mathbb{E}[E_{cut}] - \mathbb{E}[E_{cut}] \cdot p + \mathbb{E}[FP_{inter}] \tag{17}$$

$$\mathbb{E}[SHD_{inter}] = \mathbb{E}[E_{cut}](1 - \bar{p}) + \mathbb{E}[FP_{inter}] \tag{18}$$

which terminates our proof. $\qquad \square$

The intuition behind Th.4 is that the partitioning operation, besides constraining the search for a causal graph, also induces a number of false negatives, which are recovered by the merge heuristic. It is also trivial that $FP_{inter}$ can only be identical to $H_{incorrect}$ since the partitioning operation is not adding any other edge. From empirical experiments, typically LLMs excel in precision while having limited recall, implying that the term $H_{incorrect}$ tends to be small and often negligible.

We can extend it to a concentration lemma by applying Höeffding's inequality, which we recall below.

**Theorem 5** (Höeffding's Inequality (Hoeffding, 1963)). *Let $X_1, X_2, \ldots, X_n$ be independent random variables such that $a_i \le X_i \le b_i$, and let their sum be $S_i = \sum_{i=0}^{n} X_i$ and its expected value be $\mu = \mathbb{E}[S_n]$. Then, $\forall t > 0$ we have*

$$P(S_n - \mu \ge t) \le \exp\left( - \frac{2t^2}{\sum_{i=0}^{n}(b_i - a_i)^2} \right)$$

By applying the above-mentioned inequality, we have the following theorem.

**Theorem 6.** *For a confidence level $\delta \in (0, 1)$, the inter-partition $SHD_{inter}$ satisfies the following inequality with probability of at least $1 - \delta$*

$$SHD_{inter} \le \mathbb{E}[SHD_{inter}] + \sqrt{\frac{E_{cut} \cdot \ln(2/\delta)}{2}} + \sqrt{\frac{m \cdot \ln(2/\delta)}{2}}, \tag{19}$$

*where $m$ is the total number of edges proposed during the merge phase.*

*Proof.* From Th.4 we know that $SHD_{inter} = E_{cut} - H_{correct} + FP_{inter}$. To continue, we model $H_{correct}$ as a sum of $E_{cut}$ independent bernoullis, and similarly for $FP_{inter} = H_{incorrect}$, which we assume to be a sum of $m$ independent bernoullis.

First, we bound $FN_{inter}$ and $FP_{inter}$ in the same way by applying Höeffding's inequality. In both cases, being bernoulli variables, we have $a_i = 0$ and $b_i = 1$ $\forall i$.

1. $\forall t > 0$ it holds that

$$Pr(H_{correct} - \mathbb{E}[H_{correct}] \ge t) \le \exp\left( - \frac{2t^2}{E_{cut}} \right), \tag{20}$$

where, by setting the probability on the r.h.s. to $\delta/2$ yields

$$exp\left(-\frac{2t^2}{E_{cut}}\right) = \frac{\delta}{2} \implies t_{FN} = \sqrt{\frac{E_{cut} \cdot \ln(2/\delta)}{2}}. \tag{21}$$

This implies that, with probability $\geq 1 - \delta/2$ it holds that

$$H_{correct} - \mathbb{E}[H_{correct}] \leq \sqrt{\frac{E_{cut} \cdot \ln(2/\delta)}{2}} \tag{22}$$

which can be substituted into $FP_{inter} = E_{cut} - H_{correct}$, resulting into a bound for inter-partition false negatives

$$FN_{inter} \leq \mathbb{E}[FN_{inter}] + \sqrt{\frac{E_{cut} \cdot \ln(2/\delta)}{2}} \tag{23}$$

2. Similarly, for false positives we have that $\forall t > 0$ it holds that

$$Pr(FP_{inter} - \mathbb{E}[FP_{inter}] \geq t) \leq \exp\left(-\frac{2t^2}{m}\right), \tag{24}$$

which, with a procedure completely identical to the one above, we can obtain,

$$FP_{inter} \leq \mathbb{E}[FP_{inter}] + \sqrt{\frac{m \cdot \ln(2/\delta)}{2}}, \tag{25}$$

holding with probability $\geq 1 - \delta/2$.

We know that 23 and can be violated with probability $\delta/2$, and by union bound it holds that if $P(A) < \delta/2$ and $P(B) < \delta/2$, then $P(A \cup B) \leq P(A) + P(B) < \delta/2 + \delta/2 = \delta$. Therefore we have that with probability $\delta$

$$FP_{inter} + FN_{inter} \leq \mathbb{E}[FP_{inter}] + \mathbb{E}[FN_{inter}] + \sqrt{\frac{E_{cut} \cdot \ln(2/\delta)}{2}} + \sqrt{\frac{m \cdot \ln(2/\delta)}{2}} \tag{26}$$

$$SHD_{inter} \leq \mathbb{E}[SHD_{inter}] + \sqrt{\frac{E_{cut} \cdot \ln(2/\delta)}{2}} + \sqrt{\frac{m \cdot \ln(2/\delta)}{2}} \tag{27}$$

which concludes our proof. $\square$

Further, in real-world and large-scale settings where no ground-truth is available, this lower bound can be estimated by drawing a subset of the inter-partition edges, and checking with a domain expert how many were correctly estimated. With this procedure, it is possible to draw an estimate of $FP_{inter}$, $FN_{inter}$ and $E_{cut}$ to evaluate the real performance of CAST.

**Intra-Partition Analysis** We hereby show that under the assumptions of sound and completeness, errors during the Conquer phase appear mainly as False Positives.

In case a partitioning is incorrect, some d-separating sets might end being scattered across partitions, making it impossible to find those sets *within a partition*. This results in FCI being impeded to remove certain edges, which we call *spurious adjacencies* (SA).

**Definition 4** (Spurious Adjacency). *Let $\boldsymbol{P}_i$ be a subset of $\boldsymbol{V}$, and let $\mathcal{G}$ be a graph such that $(u,v) \notin \boldsymbol{E}$ with $u,v \in \boldsymbol{V}$. We define a **Spurious Adjacency** between $u$ and $v$ as an additional edge appearing when all possible conditioning sets are in $\boldsymbol{V}$, but are not in $\boldsymbol{P}_i$. Formally, for a partition $\boldsymbol{P}_i$*

$$SA_i = \sum SA \tag{28}$$

*Further, for a whole partitioning $\boldsymbol{P}$, we denote the number of all spurious adjacencies that occurred as*

$$SA(P, \mathcal{G}) = \sum SA(\boldsymbol{P}_i, \mathcal{G}_i) \tag{29}$$

In other words, the performance of FCI during the conquer step is hindered by the quality of the partitioning operation, as a bad partitioning might result in breaking the d-separating set across partitions (a d-separation failure), which results in false positives i.e. Spurious Adjacencies.

From this observation, we can derive an hard lower error-bound for the Conquer step

**Theorem 7** (Intra-Partition Expected error). *Given a causal discovery algorithm $\mathcal{F}$ which is sound and complete given the conditional independencies observable within a partition, we have*

$$SHD_{intra} = SA + FP_{LLM} + FN_{intra} \tag{30}$$

*where $FP_{LLM}$ are false positives derived from the LLM. In expectation,*

$$\mathbb{E}[SHD_{intra}] = SA + \mathbb{E}[FP_{LLM}] + \mathbb{E}[FN_{intra}]. \tag{31}$$

*Proof.* The expected intra-partition SHD can be decomposed into the sum of $\mathbb{E}[SHD_{intra}] = \mathbb{E}[FP_{intra}] + \mathbb{E}[FN_{intra}]$.

We study first the $FP_{intra}$ term. Within a partition, if $\mathcal{F}$ is sound and complete, an edge is kept if and only if no d-separating set $\mathbf{Z} \in \mathbf{P}_i$ s.t. $u \perp v | \mathbf{Z}$ is found. At the same time, in every occasion where a d-separating set is present, the edge will be removed. Therefore, $FP_{intra}$ can either be caused by:

1. The conditioning set $\mathbf{Z}$ not being present within the partition due to a bad partitioning. This produces a spurious adjacencies.

2. The LLM agents $\mathcal{C}_{hyp}$ and $\mathcal{C}_{critic}$ choosing a wrong prior constraint. This results in a $FP_{LLM}$ term.

Therefore we have $FP_{intra} = SA + \mathbb{E}[FP_{LLM}]$. For the By summing those terms, we get

$$SHD_{intra} = SA + FP_{LLM} + FN_{intra}, \tag{32}$$

and by linearity of expectation, we also have

$$\mathbb{E}[SHD_{intra}] = SA + \mathbb{E}[FP_{LLM}] + \mathbb{E}[FN_{intra}], \tag{33}$$

which terminates our proof. $\square$

The precision of the LLMs is crucial for achieving good intra-partition performance. From our experiments, the intra-partition precision of LLMs it typically very close to 1, which is enabled by the critic mechanism that helps filtering out unwanted edges.

**Theorem 8** (Upper Bound for $SHD_{intra}$). *The following holds with probability $1 - \delta$*

$$SHD_{intra} < SA + \mathbb{E}[FP_{LLM}] + FN_{intra} + \sqrt{\frac{(N_{max} - N_{edges})\ln(1/\delta)}{2}} \tag{34}$$

*Proof.* Given the maximum number of edges that the ground truth graph can have $N_{max}$, and the actual number of edges $|E| = N_{edges}$, we see that the difference $N_{max} - N_{edges}$ represents the maximum number of false positives possible, i.e. $N_{max} - N_{edges} \leq FP$. We model every potential false positive as an independent bernoulli trial that models the probability of the LLM inserting an edges. With this modeling assumption, it follows that $FP_{LLM}$ is the sum of $N_{max} - N_{edges}$ bernoulli variables, and its expected value

is $\mathbb{E}[FP_{LLM}] = \sum_{e=0}^{N_{max}-N_{edges}} p_e$. Similarly to other theorems, we can apply Höeffding's inequality to bound $FP_{LLM}$ from above:

$$Pr(FP_{LLM} - \mathbb{E}[FP_{LLM}] \geq t) \leq \exp\Big(-\frac{2t^2}{N_{max}-N_{edges}}\Big), \tag{35}$$

where, by setting the probability on the r.h.s. to $\delta$ yields

$$exp\Big(-\frac{2t^2}{N_{max}-N_{edges}}\Big) = \delta \implies t_{FP} = \sqrt{\frac{(N_{max}-N_{edges})\cdot\ln(1/\delta)}{2}}. \tag{36}$$

It follows that $FP_{LLM} \geq \mathbb{E}[FP_{LLM}] + t_{FP}$ holds with probability $\leq \delta$. Inversely, with probability of at least $1-\delta$ we have that

$$FP_{LLM} \geq \mathbb{E}[FP_{LLM}] + t_{FP} \tag{37}$$

which can be substituted into $SHD_{intra} = SA + FP_{LLM} + FN_{intra}$ to obtain

$$SHD_{intra} = SA + FP_{LLM} + FN_{intra} \leq SA + \mathbb{E}[FP_{LLM}] + t_{FP} + FN_{intra} \tag{38}$$

$$\square$$

We remark that this identity is improved further in the asymptotic limit as FCI removes all false negatives in the infinite-data limit.

**Theorem 9.** *Theorem 8 in the infinite-data limit becomes*

$$SHD_{intra} =\leq SA + \mathbb{E}[FP_{LLM}] + +\sqrt{\frac{(N_{max}-N_{edges})\ln(1/\delta)}{2}} \tag{39}$$

*Proof.* For $n \to +\infty$ and under the causal faithfulness assumption, we know that FCI is capable to remove all false negatives, i.e. $FN_{intra} \to 0^+$, from which the result. $\square$

**General Results** After analyzing what happens within partitions, and between partitions, we aggregate all the past results in order to obtain general estimate on the most important drivers for CAST's performance.

**Theorem 10.** *Let $\mathbf{P}$ be a partitioning induced by $\mathcal{D}_{hyp}$ and $\mathcal{D}_{critic}$. Define $SA$ as the number of spurious adjacencies, $E_{cut}$ and $N_{NotIntra}$ the number of inter-partition edges present resp. absent in the ground truth. Let $FPR_{\mathcal{C}}$ denote the false-positive rate of Conquer agents, and let $Rec_{\mathcal{M}}$ and $FDR_{\mathcal{M}}$ denote respectively the recall and false-discovery rate of Merge agents. Then, as $n \to \infty$, the expected total structural Hamming distance satisfies*

$$\mathbb{E}[SHD_{total}] \approx (SA + N_{NotIntra} \cdot FPR_{\mathcal{C}}) + (E_{cut}(1 - Rec_{\mathcal{M}}) + m \cdot FDR_{\mathcal{M}}). \tag{40}$$

*Proof.* We start from the following decomposition

$$\mathbb{E}[SHD_{total}] = \mathbb{E}[SHD_{infra}] + \mathbb{E}[SHD_{inter}] \tag{41}$$

From previous theorems we know that, in the infinite data limit and under faithfulness, $\mathbb{E}[SHD_{infra}] \to \mathbb{E}[FP_{infra}]$. Further, we know that previous analysis that $\mathbb{E}[FP_{infra}] = SA + \mathbb{E}[FP_{LLM}]$. How are $FP_{infra}$ generated? Given how FCI works, in the asymptotic limit those can only be generated by the LLM inserting them as prior constraints. All other possible false-positives can only be spurious adjacencies, since all other kinds of false-positives are asymptotically removed by FCI. Therefore, we have

$$\mathbb{E}[FPR_{LLM}] = \frac{\mathbb{E}[FP_{LLM}]}{N_{NotIntra}} \implies \mathbb{E}[FP_{LLM}] = N_{NotIntra} \cdot \mathbb{E}[FPR_{LLM}]. \tag{42}$$

For the inter-partition SHD, we can conduct a similar analysis. We start with the decomposition $\mathbb{E}[SHD_{inter}] = \mathbb{E}[FN_{inter}] + \mathbb{E}[FP_{inter}]$. We observe that the number of false negatives corresponds to the number of edges seved by the partitioning minus the ones recovered by the heuristic, so we have

$$\mathbb{E}[FN_{inter}] = E_{cut} - \mathbb{E}[H_{correct}]. \tag{43}$$

The recall of the merge heuristic is defined as $\mathbb{E}[Rec_{LLM}] = \frac{\mathbb{E}[H_{correct}]}{E_{cut}}$ which implies that $\mathbb{E}[H_{correct}] = E_{cut} \cdot Rec_{LLM}$. By sustitution, we get $\mathbb{E}[FN_{inter}] = E_{cut}(1 - Rec_{LLM})$. For $FP_{inter}$, it can only be identical to $H_{incorrect}$ since only the Merge heuristic can add new edges during the merge step. We can then develop an expression for $FDR_{\mathcal{M}}$ as

$$\mathbb{E}[FP_{inter}] = m \cdot FDR_{\mathcal{M}} \tag{44}$$

where $m$ is the number of edges proposed by the heuristic. By aggregating all those expression together, we get our general result,

$$\mathbb{E}[SHD_{total}] \approx (SA + N_{NotIntra} \cdot FPR_{\mathcal{C}}) + (E_{cut}(1 - Rec_{\mathcal{M}}) + m \cdot FDR_{\mathcal{M}}). \tag{45}$$

where we let $FPR_{LLM} = FPR_{\mathcal{C}}$ and $Rec_{LLM} = Rec_{\mathcal{M}}$ to highlight the importance of conquer and merge phases. $\square$

In essence, we can now see more clearly how the total SHD depends on: 1) The quality of the partitioning, as the number of spurious adjacencies might increase under bad partitioning, 2) The quality of the heuristic, as merge agents act as a safety-net to recover edges lost due to partitioning, and 3) The quality of the LLM's prior constraints within each partition.

## A.3 CONFLICT MANAGEMENT

Conflicts might arise when, after performing causal discovery on two partitions that are overlapping, we observe different edges in different partitions.

From a theoretical standpoint, the exact/most rigorous way of managing conflicts is to search for conditioning sets on the merged graph, but that is not applicable to large-size graphs due to computational reasons. Therefore, we need a tractable criterion, and for CausalSteward we adopted what follows.

Let $u, v \in \mathbf{P}_1 \subseteq \mathbf{V}$ and $u, v \in \mathbf{P}_2 \subseteq \mathbf{V}$, i.e. both nodes belong to the overlap $\mathbf{P}_1 \cap \mathbf{P}_2$. Further, denote with $G_1$ the graph derived from $\mathbf{P}_1$ and with $G_2$ the graph derived from $\mathbf{P}_2$.

We identify 3 cases:

1. An edge (any type) is present in $G_1$ but not in $G_2$: In this case, it means there exist a conditioning set s.t. $u \perp v \mid \mathbf{Z}$ with $\mathbf{Z} \in \mathbf{P}_2$ and $\mathbf{Z} \notin \mathbf{P}_1$. Since $\mathbf{Z} \subset \mathbf{P}_2 \subset \mathbf{V} \implies u \perp v \mid \mathbf{Z}$ with $u, v, \mathbf{Z} \in \mathbf{V} \implies$ **No edge between $u$ and $v$**.

2. For all other cases, we "overlap" the causal relationship: If the edge $u \to v$ is present in $G_1$, but the opposite one $v \to u$ is also present in $G_2$, we insert a bi-directed edge. If the edge $u \to v$ is present in $G_1$, but the edge $v \circ - \circ u$ is present in $G_2$, we insert the $u \circ - > v$ edge.

Also other available methods leveraging Divide&Conquer adopt approximations to manage conflicts tractably: For example, DCDILP (Dong et al. 2025) manages conflicts via Linear Programming, and other methods maximize a BIC score greedily.

## B ADDITIONAL RESULTS

Hereby, we present a more complete exposition of our results which also includes additional metrics.

## B.1 CASE-STUDY ON EARTHQUAKES DATASET

The Earthquakes dataset contains 7 variables that describe occurences of earthquakes by date, geographic location, and power. It contains the variables $\mathbf{V} = \{richter, deaths, day, month, year, area, region\}$.

Given the small graph size, CAST partitioned the nodes into three groups: $\mathbf{P}_1 = \{richter, deaths\}$, $\mathbf{P}_2 = \{day, month, year\}$, and $\mathbf{P}_3 = \{area, region\}$.

We observed that, during the explain phase, it understood correctly the variable meaning without the need of the domain expert. During the conquer phase:

- On the first cluster, LLMs correctly establish that richter → deaths.
- On the second, the hypothesis agent suggests year → month → day, which is discarded by the critic agent, showing a nice corrective mechanism.
- On the third, it correctly suggests that region → area, as an area is contained within a region, and some regions are more susceptible to earthquakes.

During the merge-phase, CAST reflects on possible connections between clusters. Here is an extract:

> **Conquer Agent on CausalChambers**
>
> "[...] the time an event occurs does not cause it to occur in a particular area – and intervening on the date would not change the location of the earthquake. [...] The location (area, region) doesn't "cause" the earthquake's measurable magnitude or human toll. Instead, geographic factors might correlate with differences (for example, construction practices or terrain might affect death tolls) but they do not cause the energy release (richter) of the earthquake. So edges like (area, richter) or (region, deaths) are poorly characterized as direct causal effects."

Overall, we have more meaningful causal relations compared to prior works. CAST is more conservative, and removes connections when uncertain.

## B.2 CAUSALMAN SMALL

| Dataset | CausalMan Small | | | | |
|---|---|---|---|---|---|
| Method | F1 ↑ | Prec.↑ | Recall ↑ | SHD ↓ | Runtime ↓ |
| FCI | 0.028(0.012) | 0.021(0.005) | 0.043(0.012) | 221.4(3.36) | 9.37(0.21) |
| XGES | 0.056(0.009) | 0.037(0.006) | 0.114(0.159) | 448.8(14.89) | 25,173.80(7,813.26) |
| GranDAG | 0.002(0.003) | 0.018(0.040) | 0.002(0.004) | 113.4(2.51) | 333.43(5.22) |
| CausalCopilot | 0.078(0.012) | 0.260(0.100) | 0.048(0.012) | 117.4(6.4) | - |
| BOSS | 0.054(0.004) | 0.029(0.002) | - | 864.20(18.60) | - |
| LLM-BFS(o3-mini) | 0.283(0.000) | **1.000(0.000)** | - | 315.2(0.0) | - |
| Pairwise(Qwen3-14B) | 0.153(0.008) | 0.093(0.005) | - | 583.50(11.1) | - |
| Pairwise(o3-mini) | 0.126(0.004) | 0.074(0.003) | 0.421(0.017) | 710(6.2) | - |
| o3-mini | | | | | |
| CAST (HITL+RAG) | 0.303(0.076) | 0.454(0.123) | 0.228(0.055) | 118.4(11.35) | 316.65(40.26) |
| CAST (HITL) | 0.304(0.062) | 0.485(0.101) | 0.224(0.053) | 112.2(10.40) | **254.66(18.54)** |
| CAST (RAG) | 0.235(0.093) | 0.373(0.118) | 0.178(0.080) | 121.4(6.34) | 455.31(115.65) |
| gpt 4o-mini | | | | | |
| CAST (HITL+RAG) | 0.142(0.027) | 0.217(0.063) | 0.105(0.018) | 140.4(13.00) | 1,630.19(614.08) |
| CAST (HITL) | 0.139(0.056) | 0.326(0.149) | 0.096(0.042) | 126.8(15.93) | 1,456.33(580.47) |
| CAST (RAG) | 0.113(0.039) | 0.169(0.047) | 0.089(0.038) | 145.0(14.35) | 913.15(231.66) |
| Qwen 3 14B | | | | | |
| CAST (HITL+RAG) | 0.424(0.082) | 0.724(0.169) | 0.307(0.070) | 103.00(10.07) | 1,064.03(231.38) |
| CAST (HITL) | **0.483(0.062)** | 0.608(0.069) | **0.402(0.059)** | **102.4(17.27)** | 1,856.65(477.45) |
| CAST (RAG) | 0.426(0.033) | 0.656(0.176) | 0.335(0.072) | 111.4(14.44) | 944.92(164.78) |

Table 2: **Results on CausalMan Small.** Different CAST variants largely outperform other baselines on Small. CausalCopilot executed FCI for Small.

| CausalMan Small | | | | |
|---|---|---|---|---|
| | LLM | #Tokens | #HITL Calls | #RAG Calls |
| HITL+RAG | Qwen3-14B | 161,500.8(38,975.15) | 3.6(2.88) | 0 |
| | o3-mini | 148,650.8 (23,167.15 ) | 0 | 0 |
| | GPT 4o-mini | 692,451.4(161,356.91 ) | 1.60(2.07) | 85.20(21.67) |
| HITL | Qwen3-14B | 141,070.2(15,844.65) | 5.6(3.21) | - |
| | o3-mini | 132,012.6(15,839.26) | 0 | - |
| | GPT 4o-mini | 126,612.8(43,272.24) | 61.00(29.86) | - |
| RAG | Qwen3-14B | 146,948.4 (23,176.93 ) | - | 2.00(1.58) |
| | o3-mini | 161,193.0(37,013.54 ) | - | 0 |
| | GPT 4o-mini | 440,740.2(115,599.65 ) | - | 69.4(16.38) |

Table 3: **CAST**. Ablation for different LLMs on CausalMan Small.

## B.3 CAUSALMAN MEDIUM

| Dataset | CausalMan Medium | | | | |
|---|---|---|---|---|---|
| Method | F1 ↑ | Prec.↑ | Recall ↑ | SHD ↓ | Runtime ↓ |
| FCI | Intractable | | | | |
| XGES | 0.014(0) | 0.211(0) | 0.007() | 547(0) | - |
| GranDAG | 0(0) | 0(0) | 0(0) | **544.4(5.77)** | 1061.17(18.91) |
| CausalCopilot | 0.014(0) | 0.211(0) | 0.007(0) | 547(0) | - |
| BOSS | 0.057(0.005) | 0.030(0.002) | - | 1,817.4(48.3) | - |
| o3-mini | | | | | |
| CAST (HITL+RAG) | 0.141(0.064) | 0.357(0.119) | 0.089(0.043) | 579.6(25.01) | 1,370.25(222.84) |
| CAST (HITL) | 0.175(0.084) | 0.499(0.127) | 0.108(0.057) | 558.8(15.498) | **1,052.11(87.07)** |
| CAST (RAG) | 0.209(0.089) | 0.518(0.201) | 0.131(0.057) | 571.6(45.845) | 1,321.15(264.94) |
| gpt 4o-mini | | | | | |
| CAST (HITL+RAG) | 0.086(0.042) | 0.271(0.113) | 0.051(0.023) | 568.2(20.71) | 2,488.79(2234.79) |
| CAST (HITL) | 0.103(0.039) | 0.392(0.083) | 0.056(0.023) | 547.0(6.89) | 1,929.17(859.52) |
| CAST (RAG) | 0.113(0.049) | 0.296(0.094) | 0.072(0.035) | 594.2(81.69) | 1,241.17(627.636) |
| Qwen 3 14B | | | | | |
| CAST (HITL+RAG) | **0.274(0.052)** | **0.656(0.110)** | 0.175(0.036) | 558.0(21.86) | 4,179.19(327.94) |
| CAST (HITL) | 0.266(0.019) | 0.508(0.222) | **0.191(0.021)** | 610.80(68.00) | 5,905.02(1139.03) |
| CAST (RAG) | 0.211(0.106) | 0.485(0.129) | 0.139(0.082) | 568.4(23.64) | 4,901.53(518.27) |

Table 4: **Results on CausalMan Medium.** Different CAST variants achieve strong competitive results on Medium. CausalCopilot executed XGES for Medium. FCI could not run for Medium.

| | CausalMan Medium | | | |
|---|---|---|---|---|
| | LLM | #Tokens | #HITL Calls | #RAG Calls |
| HITL+RAG | Qwen3-14B | 585,953.4(36,933.9) | 10.6(4.8) | 2.0(0.71) |
| | o3-mini | 563,568.2(81,608.5) | 0 | 0 |
| | GPT 4o-mini | 588,292.4(280,373.52) | 8.60(11.57) | 58.00(17.82) |
| HITL | Qwen3-14B | 525,285(115,550.9) | 15.2(5.39) | - |
| | o3-mini | 482,419.2(51,456.2 ) | 0.87(0.52) | - |
| | GPT 4o-mini | 218,590.4(69,261.0) | 40.07(17.74) | - |
| RAG | Qwen3-14B | 668,371.4(61,898.7) | - | 13.2(2.91) |
| | o3-mini | 544,760.0(96,936.4) | - | 0 |
| | GPT 4o-mini | 749,905.2(356,113.73 ) | - | 85.2(42.05) |

Table 5: **CAST**. Ablation for different LLMs on CausalMan Medium.

## B.4 Neuropathic Pain Dataset

| Dataset | Neuropathic Pain Dataset | | | | |
|---|---|---|---|---|---|
| Method | F1 ↑ | Prec.↑ | Recall ↑ | SHD ↓ | Runtime ↓ |
| FCI | Intractable | | | | |
| XGES | 0.309(0.003) | 0.442(0.005) | 0.238(0.003) | 786.40(5.177) | 520.24(6.57) |
| GranDAG | 0.004(0.000) | 0.009(0.000) | 0.003(0.000) | 993.60(2.88) | **30.59(0.182)** |
| CausalCopilot | **0.334(0)** | 0.457(0) | **0.264(0)** | 778(0) | 769(12) |
| o3-mini | | | | | |
| CAST (HITL+RAG) | 0.104(0.032) | 0.572(0.183) | 0.057(0.018) | 758.80(26.84) | 1,663(1,283.45) |
| CAST (HITL) | 0.125(0.002) | **0.674(0.041)** | 0.069(0.002) | **742.67(4.04)** | 801.87(13.69) |
| CAST (RAG) | 0.095(0.038) | 0.552(0.447) | 0.053(0.022) | 762.00(21.55) | 790.77(137.81) |
| gpt 4o-mini | | | | | |
| CAST (HITL+RAG) | 0.030(0.019) | 0.145(0.090) | 0.017(0.010) | 831.40(13.76) | 4,115.49(2,306.32) |
| CAST (HITL) | 0.027(0.026) | 0.235(0.249) | 0.014(0.014) | 797.67(27.46) | 4,763.00(1,949.63) |
| CAST (RAG) | 0.043(0.031) | 0.151(0.091) | 0.026(0.018) | 850.20(28.99) | 3,543.94(2,202.961) |
| Qwen3-14B | | | | | |
| CAST (HITL+RAG) | 0.013(0.002) | 0.152(0.008) | 0.007(0.001) | 795.00(1.41) | 3,250.408(1,412.83) |
| CAST (HITL) | 0.012(0.002) | 0.126(0.046) | 0.006(0.001) | 802.67(13.32) | 6,214.86(5,231.01) |
| CAST (RAG) | 0.064(0.051) | 0.465(0.183) | 0.035(0.029) | 776.60(16.18) | 4,672.66(760.25) |

Table 6: **Results on Neuropathic Dataset.** The data is causally sufficient, binary, and the causal graph is sparse. For this reason, XGES is less in disadvantage on this dataset with respect to CausalMan. CAST is still above the baselines in terms of precision and SHD for o3-mini. When necessary, CausalCopilot was steered towards XGES, as it often preferred intractable methods such as FCI.

| Neuropathic Pain Dataset | | | | |
|---|---|---|---|---|
| | LLM | #Tokens | #HITL Calls | #RAG Calls |
| HITL+RAG | Qwen3-14B | 264,932.00(5,334.41) | 4(0.2) | 9.5(0.71) |
| | o3-mini | 379,389.2(147,545.47) | 0 | 0 |
| | GPT 4o-mini | 1,041,107.6(318,067.42) | 3.40(3.36) | 225.0(73.73) |
| HITL | Qwen3-14B | 480,190.00(372,857.35) | 23.33(33.49) | - |
| | o3-mini | 314,726.33(6,386.65) | 0 | - |
| | GPT 4o-mini | 449,445.00(241,653.30) | 148.00(101.38) | - |
| RAG | Qwen3-14B | 696,260.8(118,152.04 ) | - | 35.00(7.35) |
| | o3-mini | 330,736(44,927.24) | - | 0 |
| | GPT 4o-mini | 831,541.8(73,133.23) | - | 186.60(25.71) |

Table 7: **CAST**. Ablation for different LLMs on the Neuropathic Pain Dataset.

## B.5 Extended Discussion

**Tool usage across LLMs:** Ablating with respect to RAG and HITL, we can evaluate the impact of prior knowledge during the discovery process. However, we see that different LLMs respond differently to having

| Variant | $F_1 \uparrow$ | Precision$\uparrow$ | Recall$\uparrow$ | $SHD \downarrow$ |
|---|---|---|---|---|
| CAST (k = 5) | 0.152(0.017) | 0.409(0.125) | 0.096(0.016) | 598.67(26.25) |
| CAST (k = 10) | 0.141(0.064) | 0.357(0.119) | 0.089(0.043) | 579.60(25.01) |
| CAST (k = 20) | 0.115(0.022) | 0.445(0.015) | 0.067(0.014) | 544.00(5.65) |

Figure 6: **Ablation for partitioning hyperparameter** $k$ **on CausalMan Medium.** Experiments conducted with CAST(RAG+HITL) using o3-mini.

those new tools available. For Qwen3 and GPT4o-mini, the number of human queries is linked to an increase in performance. However, we observe that GPT 4o-mini is very prone to forget past interactions, and frequently asks multiple times for the same information. Other LLMs, in particularly o3-mini, exhibit a smaller increase in performance. Although prompts encourage human interaction, o3-mini shows a scarce amount of HITL calls, even in instances where it is necessary. From this, we deduce that the best fit for CAST are those LLMs that have good instruction-following capabilities. Finally, results show how CAST(HITL+RAG) provides a viable trade-off between automatization and human control, and is consistently better than using just RAG. Again, this holds only for those models which follow the prompted instructions.

**Ablation for Partitioning Hyperparameter** $k$**:** In CAST, the divide agents $\mathcal{D}_{hyp}$ and $\mathcal{D}_{critic}$ are queried every time a group has a number of variables bigger than $k$, which is an integer hyperparameter to choose. We remark that this partitioning is not enforced: Divide agents can choose to not partition further even if the partition-size is bigger than $k$. In Table 6, we varied this hyperparameter and observed how performance varied. We see how increasing the partition size reduces SHD but also recall, while increasing precision. Overall $F_1$ also degrades. As partitions get bigger, it becomes harder to get a good coverage, and LLMs tend to act more conservatively.

**Performance on Neuropathic-Pain diagnosis data:** Although high-dimensional, data from the Neuropathic Pain dataset is binary and causally sufficient, which results in less limitations from a causal identifiability point of view. Indeed, purely data-driven methods tend to be less in disadvantage with respect to CAST. Still, it is evident that the impact of prior knowledge permits CAST to achieve high precision and lower SHD. We still observe, again, that other LLM baselines could not run due to slow inference (LLM-Pairwise grows $\mathcal{O}(D^2)$) and limited context window.

## C  AGENTIC RAG

One of the pillars of CAST is the capacity to dynamically query an agentic RAG system on demand. In detail, it takes the form of a tool-call, and every agent can decide when to query the RAG system. In this way, agents can use their own internal prior for trivial tasks, and retrieve further prior information when it is necessary.

### C.1  ARCHITECTURE

The system is composed by a a set of agents which, upon receiving a query $\mathcal{Q}$, are responsible for performing a web search to retrieve relevant causal knowledge.

**Web-search agents:** The system is composed by a set of agents which perform a web search. The steps are:

1. A Query agent receives the query $\mathcal{Q}$, and translates it into keywords to be used on a search engine.

2. A web-search API is executed with the keywords, and we extract the 5 best results.

3. A summary agent aggregates all results by writing a summary.

4. A reflection agent reflects on the summary and extrapolate further causal information.

5. A final summary agent provides a conclusive answer with a comprehensive explanation.

## D  DIVIDE-CONQUER-COMBINE ALGORITHM

Before detailing the logic of the process, we clarify the organization of variable partitions and their local causal graphs. Each partition is represented as $\mathbf{P}_i$, with its corresponding local causal graph defined as

$$\mathcal{G}_i := \mathcal{G}(\mathbf{P}_i, \mathbf{E}_i),$$

where $\mathbf{E}_i$ is the edge list for that partition.

Partitions are stored as nodes within a Partition Tree, denoted by T, whose structure has the following key aspects.

**Attributes of each partition node:**

- **Variable list** ($\mathbf{P}_i$): the subset of variables in the partition.
- **Edge list** ($\mathbf{E}_i$): the edges defining the local causal graph.
- **Description**: specific information about the partition, used together with the general description $I$ to write prompts $\mathbf{p}$ for the agents.
- **Solved flag** (`solved`): a boolean indicating whether the partition has been processed by the Conquer or Merge operations.

**Functions of the Partition Tree:**

- **Remove node:** deletes a partition node from T.
- **Retrieve children:** obtains the list of child partitions for a given node. For example, for a partition $\mathbf{P}_i$, its subpartitions $\{\mathbf{P}_{i,1}, \ldots, \mathbf{P}_{i,N}\}$ are stored as its children.
- **Find unsolved leaves:** identifies the leaf nodes (nodes without children) that have not yet been marked as `solved`.
- **Find ready parents:** identifies parent nodes for which all children have been marked as `solved`.

The root node of T stores the full variable set $\mathbf{V}$.

## E  BENCHMARKING SETTING

In this section we provide additional details on the benchmarking setting, including details on the implementation and how each dataset has been pre-processed.

**Hardware:**  Experiments for CAST using GPT 4o-mini and o3-mini were performed through Azure OpenAI. For Qwen3-14B, experiments were performed by using an A100 GPU (80GB VRAM) and an AMD EPYC 7643 CPU. Identically, also GranDAG used the same GPU. XGES instead has been run as a CPU-only job.

---

**Algorithm 1** Agentic Divide Operation

---

**Input:** partition $\mathbf{P}_i \subset \mathbf{V}$, partition tree T
**Output:** updated partition tree T
**function** DIVIDE($\mathbf{P}_i$, T)
    **if** $|\mathbf{P}_i| < \mathcal{K}$ **then**
        **return**                               ▷ Partition size is acceptable
    **else**
        $\{\mathbf{P}_{i,1}, \ldots, \mathbf{P}_{i,N}\}_{hyp} \leftarrow \mathcal{D}_{hyp}(\mathbf{P}_i, I, \mathbf{p}_{div.hyp})$         ▷ If not, invoke agents again
        $\{\mathbf{P}_{i,1}, \ldots, \mathbf{P}_{i,N}\}_{critic} \leftarrow \mathcal{D}_{critic}(\{\mathbf{P}_{i,1}, \ldots, \mathbf{P}_{i,N}\}_{hyp}, I, \mathbf{p}_{div.critic})$
        **if** $N > 1$ **then**            ▷ Agents do not always propose subpartitions
            Add $\{\mathbf{P}_{i,1}, \ldots, \mathbf{P}_{i,N}\}_{critic}$ as children of $\mathbf{P}_i$ in T
            **for all** $\mathbf{P}_{i,j} \in \{\mathbf{P}_{i,1}, \ldots, \mathbf{P}_{i,N}\}_{critic}$ **do**
                DIVIDE($\mathbf{P}_{i,j}$, T)         ▷ Apply recursively to subpartitions
            **end for**
        **end if**
    **end if**
    **return** T
**end function**

---

**Algorithm 2** Key Operations on Partition Tree data structure

---

**function** REMOVENODE($\mathbf{P_i}$, T)           ▷ Remove partition node $\mathbf{P_i}$, pruning T
**end function**
**function** CHILDRENOF($\mathbf{P_i}$, T)       ▷ Find all children nodes, i.e. sub-partitions, of $\mathbf{P}_i$ in T
**end function**
**function** FINDUNSOLVEDLEAVES(T)       ▷ Find all leaf nodes in T not marked "solved"
**end function**
**function** FINDREADYPARENTS(T)    ▷ Find all nodes in T whose children are all marked "solved"
**end function**

---

**Algorithm 3** Agentic Conquer Operation

---

**Input:** leaf partition node $\mathbf{P}_i \subset \mathbf{V}$, partition tree T, extended description $I$ made by explainer agent $\mathcal{E}$.
**Output:** T with $\mathbf{P}_i$ "solved", i.e. with a local causal graph $\mathcal{G}_i$
**function** CONQUER($P_i$, T)
    $\mathcal{G}_i^{hyp} \leftarrow \mathcal{C}_{hyp}(\mathbf{P}_i, I, \mathbf{p}_{hyp})$           ▷ Local graph $\mathcal{G}_i := \mathcal{G}(\mathbf{P}_i, \mathbf{E}_i)$ hypothesis
    $\mathcal{G}_i^{critic} \leftarrow \mathcal{C}_{hyp}(\mathcal{G}_i^{hyp}, I, \mathbf{p}_{hyp})$           ▷ Local graph revision
    $\tilde{\mathcal{G}}_i^{critic} \leftarrow \mathcal{F}(\mathtt{D}_i, \mathcal{G}_i^{critic})$      ▷ Run FCI on data subset $\mathtt{D}_i$, with $\mathcal{G}_i$ as prior knowledge
    Store $\tilde{\mathcal{G}}_i^{critic}$ in T
    Mark $\mathbf{P}_i$ with a "solved" flag
    **return** T
**end function**

---

---

**Algorithm 4** Agentic Combine Operation

---

**Input:** parent partition node $\mathbf{P}_i \subset \mathbf{V}$, partition tree T
**Output:** T with $\mathbf{P}_i$ "solved", i.e. with a local causal graph $\mathcal{G}_i$
**function** COMBINE($\mathbf{P}_i$, T)
    $\mathbf{C} \leftarrow$ CHILDRENOF($\mathbf{P_i}$, T)                         ▷ Defined in 2
    Initialize $\mathbf{E}_i \leftarrow \emptyset$
    **for all** $\mathbf{P}_{i,j}$ in $\mathbf{C}$ **do**
        $\mathbf{E}_i \leftarrow \mathbf{E}_i \cup \mathbf{E}_{i,j}$               ▷ Unite edge lists of local graphs
        REMOVENODE($\mathbf{P}_{i,j}$, T)                  ▷ Defined in 2
    **end for**
    $\mathcal{G}_{comb}^{hyp} \leftarrow \mathcal{M}_{hyp}(\mathcal{G}_1, \ldots, \mathcal{G}_M, I, \mathbf{p}_{comb})$      ▷ Hypothesis for bridging edges
    $\mathcal{G}_{comb}^{critic} \leftarrow \mathcal{M}_{hyp}(\mathcal{G}_i^{hyp}, I, \mathbf{p}_{comb})$           ▷ Revised bridging edges
    $\mathcal{G}_i \leftarrow \mathcal{G}_{comb}^{critic} \cup \mathcal{G}(\mathbf{P_i}, \mathbf{E}_i)$        ▷ Unite bridging and local edges
    Store $\mathcal{G}_i$ in T
    Mark $\mathbf{P}_i$ as "solved" in T
    **return** T
**end function**

---

**Algorithm 5** Agentic Explain, Divide, Conquer and Combine Phases

---

**Input:** Data D with variable set $\mathbf{V}$, integer threshold $\mathcal{K}$
**Output:** Causal graph $\mathcal{G}(\mathbf{V}, \mathbf{E})$
$I \leftarrow \mathcal{E}(\mathsf{D}_{labels}, \mathbf{p}_{explain})$                     ▷ Generate description $I$
Initialize T with root node $\mathbf{V}$
$\{\mathbf{P}_1, \ldots, \mathbf{P}_N\}_{hyp} \leftarrow \mathcal{D}_{hyp}(V, I, \mathbf{p}_{div.hyp})$      ▷ Hypothesis partition of $\mathbf{V}$
$\{\mathbf{P}_1, \ldots, \mathbf{P}_N\}_{critic} \leftarrow \mathcal{D}_{critic}(\{\mathbf{P}_1, \ldots, \mathbf{P}_N\}, I, \mathbf{p}_{div.critic})$    ▷ Revised partition of $\mathbf{V}$
Add $\{\mathbf{P}_1, \ldots, \mathbf{P}_N\}$ as children of $\mathbf{V}$ in T        ▷ Store partitions in T
**for** $\mathbf{P}_i$ in $\{\mathbf{P}_1, \ldots, \mathbf{P}_N\}$ **do**            ▷ Repeat recursively
    DIVIDE($\mathbf{P}_i$, T)                          ▷ Defined in 1
**end for**
**while** $V$ is not marked "solved" in T **do**
    $\mathbf{L} \leftarrow$ FINDUNSOLVEDLEAVES(T)             ▷ Defined in 2
    **while** $\mathbf{L} \neq \emptyset$ **do**
        **for all** $\mathbf{P}_{leaf}$ in $\mathbf{L}$ **do**        ▷ Solve the leaf nodes of T
            CONQUER($\mathbf{P}_{leaf}$, T)             ▷ Defined in 3
        **end for**
        $\mathbf{R} \leftarrow$ FINDREADYPARENTS(T)            ▷ Defined in 2
        **while** $\mathbf{R} \neq \emptyset$ **do**
            **for all** $P_{parent}$ in $\mathbf{R}$ **do**    ▷ Combine the local graphs of solved partitions
                COMBINE($P_{parent}$, T)           ▷ Defined in 4
            **end for**
            $\mathbf{R} \leftarrow$ FINDREADYPARENTS(T)
        **end while**
        $\mathbf{L} \leftarrow$ FINDUNSOLVEDLEAVES(T)
    **end while**
**end while**
**return** $\mathcal{G}(\mathbf{V}, \mathbf{E})$ from T

---

### E.1 NEUROPATHIC PAIN DATASET:

We perform our experiments on the Neuropathic pain dataset (222 nodes). All variables are binary and we do not perform any normalization.

We remark that the data-driven FCI algorithm within the Conquer phase is executed by extracting the relevant columns from the dataset (related to the nodes in the sub-graph to refine). However, this dataset contains a number of constant columns, therefore we added a small gaussian noise $\epsilon \sim \mathcal{N}(0, \sigma)$ with $\sigma = 0.0001$ to the dataset, with the goal of maintaining the correlation matrix always non-singular. This is necessary, as the $\mathcal{X}^2$ conditional independence tests require it. This dataset is publicly available at Link

### E.2 CAUSALMAN DATASET:

We perform our experiments on CausalMan Small (53 nodes) and Medium (186 nodes) with 50.000 samples. Data has been normalized between -1 and 1. Categorical variables have been embedded on an equally spaced grid, still between -1 and 1. Further details on the data in the next section F.

## F MANUFACTURING DATA: CAUSALMAN DATASET

The CausaMan Dataset (Tagliapietra et al., 2025) is modeled after a real world production line performing a press-fitting process. It presents two variants, namely *CausalMan Small* and *CausalMan Medium*, which feature respectively 53 and 186 variables. From the evaluations present in Tagliapietra et al. (2025), we observe how classic data-driven methods are limited due to limitations arising from causal identifiability.

### F.1 DATA DESCRIPTION

Data from the CausalMan Dataset features different characteristics which makes it a challenging benchmark. Importantly, it exhibits a large amount of confounding effects and non-identifiable causal relationships, which bounds the efficacy of purely data-driven methods.

We recall here the main characteristics:

- **Causal Insufficiency:** A large SCM where half of the variables are hidden, implying an high level of confounding effects.
- **Hybrid Data-types:** Continuous, discrete, categoricals and binary variables are all present.
- **Non-linear Causal Mechanisms and non-additive noise models:** Causal mechanisms are often non-linear and the noise is not treated additively.
- **Discrete-to-Continuous relationships:** Relationships from discrete variables to continuous ones are present. They take the form of switching mechanisms, where the noise distribution of certain continuous nodes varies depending on the discrete values of its parents.

### F.1.1

## G PROMPTS

We present in this section all the prompt templates that have been used. We remark that every phase -**except for the explain phase**- adopts an Hypothesis-Critic approach, where an agent proposes an hypothesis for the solution, and a critic agent follows with a refined version. We recall our notation: $\mathcal{E}$ is the explainer agent, $\mathcal{D}_{hyp}$ is the Divide-Hypothesis agent, $\mathcal{D}_{critic}$ is the Divide-Critic agent, $\mathcal{C}_{hyp}$ is the Conquer-Hypothesis

agent, $\mathcal{C}_{critic}$ is the Conquer-Critic agent, $\mathcal{M}_{hyp}$ is the Merge-Hypothesis agent, and finally $\mathcal{M}_{critic}$ is the Merge-Critic agent.

We follow with every prompt template that has been used. We additionally concatenate to those prompts a number of few-shot examples on how to efficiently use the provided tool (RAG and HITL), which we omit for brevity.

---

**Explainer Agent (System Prompt)**

You are an expert in the domain domain tasked with explaining a dataset and its variables. Your goal is to provide a clear and concise description of the dataset, including the meaning of each variable and their potential relationships. You will receive a list of variables and, optionally, a dataset description. Utilize the available tools to look up the meanings of variable names. Employ an iterative thought-action-observation approach to gather evidence, validate your reasoning, and refine both the description of variables.

---

**Explainer Agent (User Prompt)**

You are tasked with analyzing the following dataset:
Variable names: **{variable_names}**
Dataset description: **{dataset_description}**

Your objectives are:
(1) Refine and expand the Dataset description, if present.
(2) Provide an informative description of the dataset labels, clarifying the meaning of acronyms and context of each variable. Preserve all information in the above description, if present.
(3) Include both above results within <**general_description**></**general_description**> tags.

IMPORTANT: Web search cannot understand acronyms. If variale names are acronyms, do not use those acronyms for a web search. Instead, make the query in natural language. IMPORTANT: Variable labels in output should be the same as in the input.

---

## G.1   DIVIDE PHASE

### G.1.1   DIVIDE-HYPOTHESIS AGENT

First, the Divide hypothesis agent $\mathcal{D}_{hyp}$ received the following system prompt.

---

**Divide-Hypothesis Agent (System Prompt)**

You are an expert in the domain domain tasked with partitioning a dataset's variables into groups that might share causal relationships. Each variable represents a node in a causal graph.
Your goal is, given a list of variables, to divide them into groups.

Reason on:

- Pairs of variables that are directly causally related ("A" causes "B"). If they are, they belong in the same group.

- If they are not likely to be directly causally related, they either:
  Case 1) Belong in separate groups, meaning they are likely independent.
  Case 2) Are indirectly related through common causes ("C" causes both "A" and "B") or mediators ("A" causes "C", "C" causes "B"). In this case, you can form a group by also including the common causes or mediators.

- If two variables have semantically similar names or describe similar things, it does not necessarily mean they belong in the same group.

Additional requirements:

- The groups may overlap.

- Each variable in the dataset should be assigned to at least one group.

- You may form any number of groups.

- Include detailed information on each variable within each group.

Your process:

- Available tools should be used to gather information and validate your reasoning. Use them to to confirm variable meanings and assess relationships.

- Utilize an iterative thought-action-observation approach to gather evidence, validate your reasoning and refine your groups.

---

Following, at every iteration of the Divide phase, $\mathcal{D}_{hyp}$ adopts the following user prompt.

---

**Divide-Hypothesis Agent (User Prompt)**

You are tasked with analyzing the following dataset:
Dataset description: **{dataset_description}**
For your partitioning task, focus on the following variables: **{variable_names}**

IMPORTANT: Web search cannot understand acronyms. If variable names are acronyms, do not use those acronyms for a web search. Instead, make the query in natural language.
IMPORTANT: Variable labels in output should be the same as in the input. Each input variable should be assigned to a group. Double check as to not introduce new variables by accident.
IMPORTANT: Output a list of variables and their descriptions for each group. Ensure that each list is enclosed within <nodes> tags. For example: **<group><nodes>[Variable1, Variable2]</nodes><description>**Description of the group**</description></group>**

---

### G.1.2 DIVIDE-CRITIC AGENT

The Divide hypothesis agent $\mathcal{D}_{critic}$ receives the following system prompt.

---

**Divide-Critic Agent (System Prompt)**

You are an expert in the **{domain}** domain tasked with partitioning a dataset's variables into groups that might share causal relationships. Each variable represents a node in a causal graph. Your goal is to analyze groups proposed by another agent and and modify them if necessary.

Reason on:

- If a pair of variables are in the same group, they are either:
  Case 1) Directly causally related ("A" causes "B").
  Case 2) They are indirectly related through common causes ("C" causes both "A" and "B") or mediators ("A" causes "C", "C" causes "B"). In this case, the common causes or mediators should also be in the group.
- If two variables have semantically similar names or describe similar things, review whether they really belong in the same group.

Additional requirements:

- The groups may overlap.
- Each variable in the dataset should be assigned to at least one group.
- You may edit any number of groups or even create new ones.
- Include detailed information on each variable within each group, preserve any relevant information that is already present.

Your process:

- Available tools should be used to gather information and validate your reasoning. Use them to to confirm variable meanings and assess relationships.
- Utilize an iterative thought-action-observation approach to gather evidence, validate your reasoning and refine your groups.

---

Following, at every iteration of the Divide phase, $\mathcal{D}_{critic}$ adopts the following user prompt.

---

**Divide-Critic Agent (User Prompt)**

You are tasked with analyzing the following dataset:
Dataset description: **{dataset_description}**
For your partitioning task, focus on the following variables: **{variable_names}**
The groups proposed by the previous agent are: **{proposed_groups}**

IMPORTANT: Web search cannot understand acronyms. If variable names are acronyms, do not use those acronyms for a web search. Instead, make the query in natural language.
IMPORTANT: Variable labels in output should be the same as in the input. Each input variable should be assigned to a group. Double check as to not introduce new variables by accident.
IMPORTANT: Output a list of variables and their descriptions for each group. Ensure that each list is enclosed within <nodes> tags. For example: **<group><nodes>[Variable1, Variable2]</nodes><description>Description of the group</description></group>**

---

## G.2   CONQUER PHASE

### G.2.1   CONQUER-HYPOTHESIS AGENT

---

**Conquer-Hypothesis Agent (System Prompt)**

You are an expert in the **{domain}** domain tasked with identifying cause-effect relationships from data. Your goal is to construct a causal graph where nodes represent variables, and directed edges represent hypothesized cause-effect relationships. You will receive a list of variables and, optionally, a dataset description. You have access to tools to assist in gathering prior knowledge, helping you to refine your hypothesis. Use these tools iteratively to gather relevant knowledge, and refine your causal graph. Reason through each step and leveraging the tools to support your conclusions. Try to understand what each variable means. You can the tools available in case additional information is needed.

IMPORTANT: In some cases you might have access to a human expert. In that case, use it to gather contextual knowledge, and balance it between knowledge from RAG and human expert.
IMPORTANT: Do not be conservative in putting a causal relationship. It is better to put an edge in case you are in doubt even after using your tools.

---

---

**Conquer-Hypothesis Agent (User Prompt)**

Reason to get a causal graph from the following dataset.
Dataset description: **{general_description}**
You will focus on the following variables:
Variable names: **{variable_names}**
Description of variables: **{variable_description}**

Successively, you will perform the following points until you are confident about a final answer:

1. Understand the context, about what are the variables modeling at a causal level. You can use the tools to search for more information.

2. Reason on which might be the root nodes which are not influenced by other variables. Successively, reason on the relationships between those root nodes and the other child variables. Then reason between children of children, and so on...

3. Output a preliminary list of all edges that COULD POTENTIALLY be present between those variables. DO NOT BE TOO CONSERVATIVE. AN EDGE MORE IS BETTER THAN AN EDGE LESS.

4. Reflect and improve your estimate.

Finally, output a cumulative list of directed edges between the provided variables.

IMPORTANT: Make sure the the list of edges are within the <edges> tags. For example: **<edges>(Variable1, Variable2), (Variable2, Variable3)</edges>**
IMPORTANT: ABSOLUTELY RESPECT THE FORMAT. DO NOT USE ARROWS OR ANYTHING SIMILAR TO REPRESENT EDGES. ONLY USE THIS FORMAT, AND USE THE COMMA TO SEPARATE VARIABLE NAMES:**<edges>(Variable1, Variable2), (Variable2, Variable3)</edges>**

---

### G.2.2 CONQUER-CRITIC AGENT

---

**Conquer-Critic Agent (System Prompt)**

You are an expert in the domain domain tasked with critically evaluating proposed causal relationships between variables in a dataset. Your goal is to analyze a hypothesized causal graph and suggest modifications, particularly focusing on identifying edges that are in the wrong anti-causal direction, or that do not hold true.

You will receive a list of variables, a dataset description, and a proposed causal graph. Use a thought-action-observation process for your reasoning. You have access to tools to assist in verifying relationships by gathering prior knowledge. Be systematic in your approach, reasoning through each step and leveraging the tools to support your conclusions.

First, assess the validity of each edge in the proposed graph and consider potential confounding relationships which might have been missed. Use the tools available to gather additional information as needed.

---

---

**Conquer-Critic Agent (User Prompt)**

Your task is to critically analyze the hypothesized causal graph based on the provided dataset. Here is the hypothesis you need to evaluate:
Dataset description: **{general_description}**
You will focus on the following variables:
Variable names: **{variable_names}**
Description of variables: **{variable_description}**
Hypothesis graph: **{causal_graph}**

You will perform the following steps until you reach a confident conclusion:

1. Evaluate the proposed relationships between the variables.

2. Assess whether any edges are in the anti-causal direction (i.e., (effect, cause) instead of (cause, effect)). Consider that intervening on the effect should not change the cause, but not the viceversa.

3. Do those causal relationships always hold, or only on some context? Think about counterfactual scenarios.

4. Utilize the available tools to gather additional information and evidence to support your analysis. If available, use the human expert for confirmation or for contextual knowledge.

Finally, provide a cumulative list of directed edges identified at each iteration for every group of variables.

IMPORTANT: Ensure that the list of edges is enclosed within <edges> tags. For example: **<edges>(Variable1, Variable2), (Variable2, Variable3)</edges>** IMPORTANT: ABSOLUTELY RESPECT THE FORMAT. DO NOT USE ARROWS OR ANYTHING SIMILAR TO REPRESENT EDGES. ONLY USE THIS FORMAT, AND USE THE COMMA TO SEPARATE VARIABLE NAMES:**<edges>(Variable1, Variable2), (Variable2, Variable3)</edges>**

## G.3 COMBINE PHASE

### G.3.1 COMBINE-HYPOTHESIS

---

**Combine-Hypothesis Agent (System Prompt)**

You are a **{domain}** domain expert responsible for evaluating causal relationships. In this context, each variable in a dataset represents a node in a causal graph. You will receive different groups of variables, each accompanied by a description. The causal relationships within each group have already been assessed. Your objective is to analyze the relationships between these groups to hypothesize potential connections bridging the graphs. The goal is to merge them into a larger causal graph. Specifically, you will propose potential directed edges (cause, effect) between variables from different groups. Utilize the available tools to verify and refine your hypotheses iteratively. Approach this task systematically, reasoning through each step and leveraging the tools to support your conclusions.

---

**Combine-Hypothesis Agent (User Prompt)**

Your task is to establish connections between variables from the following groups, which will introduce new edges in a directed causal graph. Output pairs of variables from different groups, ensuring that the relationships are consistent.

Dataset description: **{general_description}**
You will now focus on these specific groups: **{groups}**

Follow these steps until you are confident in your final answer:

1. Understand the context, about what is each group of variables modeling at a causal level. Understand how each groups is related to each other. You can use the tools to search for more information.

2. Output a preliminary list of bridging edges that COULD POTENTIALLY be present between those groups of variables. DO NOT BE TOO CONSERVATIVE. AN EDGE MORE IS BETTER THAN AN EDGE LESS.

3. Reflect and improve your estimate.

Finally, present a single list of directed edges that connect the groups.

IMPORTANT: Ensure the list of edges is enclosed within <edges> tags. For example: **<edges>(Variable1, Variable2), (Variable2, Variable3)</edges>**
IMPORTANT: ABSOLUTELY RESPECT THE FORMAT. DO NOT USE ARROWS OR ANYTHING SIMILAR TO REPRESENT EDGES. ONLY USE THIS FORMAT, AND USE THE COMMA TO SEPARATE VARIABLE NAMES:**<edges>(Variable1, Variable2), (Variable2, Variable3)</edges>**

---

### G.3.2 COMBINE-CRITIC

---

**Combine-Critic Agent (System Prompt)**

You are a **{domain}** domain expert tasked with critically analyzing the proposed connections between variable groups in a causal graph. Your role is to evaluate the validity of the directed edges (cause, effect) and suggest modifications, particularly focusing on identifying edges that are in the wrong anti-causal direction, or that do not hold true.

You will receive a list of separate groups of variables, each with its description. You will also be provided with a set of causal edges bridging those groups. You need to understand how those groups are related.

Use a thought-action-observation process for your reasoning. You have access to tools to assist in verifying relationships by gathering prior knowledge.

---

---

**Combine-Critic Agent (User Prompt)**

Your task is to critically evaluate the proposed connections between the following groups of variables.
Dataset description: **{general_description}**
You will now focus on these specific groups: **{groups}**
Proposed connections: **{group_connections}**

Follow these steps to analyze the connections:

1. Evaluate the proposed relationships between the variables.

2. Assess whether any edges are in the anti-causal direction (i.e., (effect, cause) instead of (cause, effect)). Consider that intervening on the effect should not change the cause, but not the viceversa.

3. Do those causal relationships always hold, or only on some context? Think about counter-factual scenarios.

4. Utilize the available tools to gather additional information and evidence to support your analysis. If available, use the human expert for confirmation or for contextual knowledge.

Provide a summary of your findings, including any necessary revisions to the proposed connections.

IMPORTANT: Ensure the list of edges is enclosed within <edges> tags. For example: **<edges>(Variable1, Variable2), (Variable2, Variable3)</edges>**
IMPORTANT: ABSOLUTELY RESPECT THE FORMAT. DO NOT USE ARROWS OR ANYTHING SIMILAR TO REPRESENT EDGES. ONLY USE THIS FORMAT, AND USE THE COMMA TO SEPARATE VARIABLE NAMES:**<edges>(Variable1, Variable2), (Variable2, Variable3)</edges>**

---

# H    COMPLETE END-TO-END INTERACTION

To illustrate how CAST's execution flows, we report here a complete interaction on the Earthquakes dataset. To keep it readable, we omit to repeat the system and user prompt templates (already provided in Sec.G), and only report the LLM answers, HITL Calls, and User responses. An additional end-to-end interaction on the CausalChambers dataset is provided in the supplementary materials.

**Main Takeaways**    Some interesting insights into the behavior of CAST are:

- As it is shown in the HITL calls, it is not necessary to provide "explicit" causal information of the kind "A implies B" as the interaction happens in natural language, and the extraction of causal information from user feedback is managed by the agents (Antonucci et al., 2023).

- Often, we intentionally provided high-level, partial, or even "I don't know" responses to the agents. This highlights CAST's ability to handle realistic, imperfect user input.

- You may observe occasional repeated queries. This results from the current design choice to keep agents separate and without a global shared memory. While this ensures modularity, we acknowledge that adding shared memory is a straightforward engineering optimization for future work to reduce token usage and enhance scalability further.

## H.1 EXPLAIN PHASE

Upon being given a first user-description and the dataset, the Explain phase employs an explainer agent that extends the provided information and builds a more extensive description of the whole dataset. In case of ambiguities in the variable naming, the Explainer agent will reach the Human via a call to the HITL tool.

---

**Explainer Agent (Part 1 of 2)**

### REFINED DATASET DESCRIPTION

This dataset contains records of earthquake events, including their occurrence date, location, seismic magnitude, and the number of recorded fatalities. It is intended for analyses of seismic activity and its human impact across different regions and time periods. Each record corresponds to a single earthquake, with fields that capture when it happened, where it was observed or affected, how strong it was (by a magnitude scale commonly referred to as the "Richter" scale), and how many deaths were attributed to that event.

#### VARIABLE DESCRIPTIONS

**year** Calendar year in which the earthquake occurred. Typically a four-digit integer (e.g., 1999, 2015). Useful for temporal trend analyses and aggregations.

**month** Calendar month of occurrence, usually as an integer from 1–12. Enables seasonal or monthly pattern analysis when combined with year and day.

**day** Calendar day of occurrence, usually as an integer from 1–31. Together with year and month, can be combined to form a full event date.

**region** The broader geographic region associated with the earthquake (e.g., country, multi-country region, or seismic region label). This is a categorical/location descriptor useful for grouping events and comparing regional risk and impacts.

**area** A more specific locality within the region, such as a province/state, district, city, or named seismic/epicentral area. This typically offers finer spatial granularity than region; depending on the data source, it may denote the epicenter vicinity or the administrative area most affected.

**richter** The earthquake's magnitude, labeled as "richter." In many historical datasets, this refers to magnitude reported on the Richter/local magnitude scale ($M_L$). Note that modern catalogs often report magnitudes on the moment magnitude scale ($M_w$); if the source uses $M_w$ but retains the "richter" label, values are still logarithmic magnitudes where each unit increase reflects $\sim 10\times$ increase in ground-motion amplitude and $\sim 32\times$ increase in released energy. Units: dimensionless magnitude.

**deaths** Number of recorded fatalities attributable to the earthquake event. Typically a non-negative integer. Be aware that zero can mean "no deaths recorded," while missing values may indicate "unknown." Interpretation may require scrutiny of reporting practices.

---

**Explainer Agent (Part 2 of 2)**

CONTEXT AND POTENTIAL RELATIONSHIPS

- **Temporal structure:** `year`, `month`, and `day` define the event date; analyses often aggregate by year or month to examine temporal trends in seismic activity or impact.
- **Spatial structure:** `region` and `area` are hierarchical, with `area` nested within `region` in many datasets; this supports spatial aggregation (e.g., counts by region) and finer local analyses.
- **Magnitude-impact linkage:** `richter` (magnitude) is a primary predictor of potential damage and deaths, but actual fatalities also depend on depth, proximity to population centers, building standards, and secondary hazards (e.g., tsunamis, landslides).
- **Reporting considerations:** `deaths` may be influenced by data completeness and post-event updates; magnitudes can be revised as more data becomes available.

**Preserved information:** This dataset describes data with records about earthquakes, such as their location, date, magnitude, and recorded fatalities.

---

## H.2 DIVIDE PHASE

During this phase, the Divide agent takes the description built by the explainer agent, and we first have $\mathcal{D}_{hyp}$ that proposes a possible partitioning of the variables, which is later refined by $\mathcal{D}_{critic}$. In case partitions are too large (by choosing a maximum partition size $k$), this process is iterated on the individual partitions. The result is a partition tree.

---

**Divide Hypothesis Agent (Part 1 of 2)**

VARIABLE GROUPS AND STRUCTURAL RELATIONSHIPS

**Nodes: {year, month, day}** **Temporal structure variables that jointly define the earthquake's calendar date.**

- **year:** The calendar year of occurrence (four-digit integer). It constrains valid month/day combinations and is used for trend analyses (e.g., long-term changes in seismic activity or impacts).
- **month:** Calendar month (1–12). Nested within year; together with year, it sets seasonal context (e.g., monsoon, winter), which can indirectly affect vulnerability or secondary hazards.
- **day:** Calendar day (1–31). Dependent on year and month for validity (e.g., leap years). When combined with year and month, it forms the full event date and can capture fine-grained timing effects (e.g., holidays) that may indirectly affect exposure and impact.

---

> ### Divide Hypothesis Agent (Part 2 of 2)
>
> **Nodes: {region, area}  Spatial hierarchy describing where the earthquake occurred or was most impactful.**
>
> - **region:** Broader geographic descriptor (e.g., country or multi-country seismic region). Used for coarser spatial aggregation and comparison across regions.
> - **area:** More specific locality within region (e.g., state/province, district, city, or named epicentral area). Typically nested in region; directly linked via administrative or geographic hierarchy. Differences in local population density and building stock at the area level can strongly influence observed impacts.
>
> **Nodes: {richter, deaths}  Magnitude–impact linkage.**
>
> - **richter:** Earthquake magnitude (often labeled on the "Richter"/local magnitude scale; values are logarithmic). Directly influences expected ground-motion amplitude and energy release, which in turn drive potential damage.
> - **deaths:** Recorded fatalities attributed to the event. Causally downstream of shaking and damage, but also shaped by mediators such as local vulnerability, building codes, depth, proximity to population centers, and secondary hazards. The relationship is typically positive but highly modulated by context and reporting practices.
>
> **Nodes: {year, month, day, region, area, richter, deaths}  Integrated impact pathway combining temporal, spatial, and physical drivers of human consequences.**
>
> *Pathways:*
>
> - `richter` → `deaths` (direct effect): Larger magnitudes tend to produce stronger shaking and, absent mitigation, more casualties.
> - `region/area` → `deaths` (contextual mediators): Regional/area characteristics (population density, construction practices, enforcement of building codes, terrain, tsunami/landslide susceptibility) modulate how shaking translates into fatalities.
> - `year/month/day` → `deaths` (temporal mediators): Year captures secular changes (e.g., evolving building standards, emergency response capacity, and reporting completeness). Month and day provide seasonal/contextual timing that can affect exposure (e.g., tourism seasons, weather conditions) and secondary hazards.
>
> *Variable details:*
>
> - **year:** Event year; useful for temporal trend and policy-era analyses.
> - **month:** Event month; supports seasonal pattern analysis.
> - **day:** Event day; completes the date and allows alignment to specific calendar effects.
> - **region:** Broad location for cross-region comparisons.
> - **area:** Fine-grained locality reflecting exposure heterogeneity within regions.
> - **richter:** Physical intensity driver of potential damage.
> - **deaths:** Outcome reflecting human impact, downstream of the above determinants.

---

**Divide Critic Agent (Part 1 of 2)**

### VARIABLE GROUPS AND STRUCTURAL RELATIONSHIPS

**Nodes: {year, month, day} Temporal structure variables that jointly define the earthquake's calendar date.**

- **year:** Calendar year of occurrence (four-digit integer). Constrains valid month/day combinations and supports trend analyses (e.g., long-term changes in seismic activity or impacts).
- **month:** Calendar month (1–12). Nested within year; provides seasonal context (e.g., monsoon, winter), which can indirectly affect vulnerability or secondary hazards.
- **day:** Calendar day (1–31). Dependent on year and month for validity (e.g., leap years). Combined with year and month to form the full event date; fine-grained timing can capture context (e.g., holidays) that may indirectly affect exposure and impact.

**Nodes: {region, area} Spatial hierarchy describing where the earthquake occurred or was most impactful.**

- **region:** Broader geographic descriptor (e.g., country or multi-country seismic region). Useful for coarser spatial aggregation and comparison across regions.
- **area:** More specific locality within region (e.g., state/province, district, city, or named epicentral area). Typically nested within region; local population density and building stock at the area level strongly influence observed impacts.

**Nodes: {richter, deaths} Magnitude–impact linkage.**

- **richter:** Earthquake magnitude (logarithmic scale). Directly influences expected ground-motion amplitude and energy release, driving potential damage.
- **deaths:** Recorded fatalities attributed to the event. Causally downstream of shaking and damage; the relationship with magnitude is typically positive but modulated by context (e.g., vulnerability, secondary hazards, reporting practices).

**Nodes: {year, month, day, region, area, richter, deaths} Integrated impact pathway combining temporal, spatial, and physical drivers of human consequences.**

*Pathways:*

- `richter` → `deaths` (direct effect): Larger magnitudes tend to produce stronger shaking and, absent mitigation, more casualties.
- `region`/`area` → `deaths` (contextual mediators): Regional/area characteristics (population density, construction practices, enforcement of building codes, terrain, tsunami/landslide susceptibility) modulate how shaking translates into fatalities.
- `year`/`month`/`day` → `deaths` (temporal mediators): Year captures secular changes (e.g., evolving building standards, emergency response capacity, reporting completeness). Month and day provide seasonal/contextual timing that can affect exposure (e.g., tourism seasons, weather conditions) and secondary hazards.



**Divide Critic Agent (Part 2 of 2)**

*Variable details:*

- **year:** Event year; useful for temporal trend and policy-era analyses.
- **month:** Event month; supports seasonal pattern analysis.
- **day:** Event day; completes the date and allows alignment to specific calendar effects.
- **region:** Broad location for cross-region comparisons.
- **area:** Fine-grained locality reflecting exposure heterogeneity within regions.
- **richter:** Physical intensity driver of potential damage.
- **deaths:** Outcome reflecting human impact, downstream of the above determinants.

**Nodes: {year, deaths}  Secular trends and reporting-era effects on impact.**

- **year:** Captures changes over time in building codes, urbanization, emergency response capacity, and reporting completeness.
- **deaths:** Fatalities recorded for the event; trends by year can reflect true changes in vulnerability and exposure as well as improvements in data collection.

**Nodes: {region, area, month, deaths}  Seasonal–spatial vulnerability pathway.**

- **region:** Broad spatial context affecting hazard environment (e.g., monsoon-prone regions, tsunami exposure).
- **area:** Localized exposure and vulnerability (population density, terrain, construction quality).
- **month:** Seasonal timing influencing secondary hazards (e.g., rainfall-triggered landslides, storm seasons) and exposure (tourism seasons, agricultural cycles).
- **deaths:** Fatalities as an outcome modulated by seasonal conditions and local vulnerability, even for similar magnitudes.

**Nodes: {region, area, richter, deaths}  Spatial moderation of magnitude–impact.**

- **region:** Coarse spatial unit shaping typical building practices, enforcement, and hazard profiles.
- **area:** Fine spatial unit determining proximity to population centers and local site effects (e.g., basin amplification).
- **richter:** Physical intensity driver; higher magnitudes raise the potential for severe shaking.
- **deaths:** Impact outcome; the effect of magnitude on deaths is moderated by region and area characteristics.



## H.3  CONQUER PHASE

This phase is executed for every partition that is a leaf in the partition tree (that has never been partitioned further). On each one of those partitions, the Conquer agents $\mathcal{C}_{hyp}$ and $\mathcal{C}_{critic}$ propose and refine a candidate causal graph. That causal graph is then used as a constraint for a causal discovery algorithm. During this phase, we can see (in the red and green boxes) that multiple HITL calls are made in those cases where the conquer agents are uncertain.

---

**Conquer Hypothesis Agent for variables Year/Month/Day**

STRUCTURAL EDGES

- year → day
- month → day

---

**Conquer Critic Agent for variables Year/Month/Day**

EVALUATION OF TEMPORAL CONSTRAINT EDGES

1. EVALUATION OF PROPOSED RELATIONSHIPS

**(year, day)** Valid as a structural constraint in the Gregorian calendar. The year affects whether certain days are valid (e.g., February 29 exists only in leap years), so the permissible values of day depend on year for some months.

**(month, day)** Valid as a structural constraint. The month determines the range of valid days (e.g., 30 vs. 31 days, and February's length), independent of other variables except for leap year effects that involve year.

2. ANTI-CAUSAL DIRECTION ASSESSMENT

Neither edge is in the anti-causal direction. Intervening on day does not change year or month, whereas intervening on year or month can change which day values are valid. Thus, the direction from year → day and month → day is appropriate.

3. CONTEXTUAL VALIDITY AND COUNTERFACTUALS

These relationships hold under the Gregorian calendar and similar systems where day validity depends on month and sometimes year.
**Counterfactual examples:**

- If we set month = February and intervene to change year from a leap year to a non-leap year, the set of valid day values changes (29 becomes invalid).
- If we set year fixed and intervene to change month from April to May, the valid day set changes (30 vs. 31 days).

Outside this calendar context (e.g., non-Gregorian calendars), the specific constraints might differ. However, the dataset description implies standard calendar usage.

4. ADDITIONAL CONSIDERATIONS

No additional confounders among year, month, and day are necessary for validity constraints. The relationships are deterministic constraints rather than causal mechanisms in the physical sense, but they are directionally correct for data-generating validity.

FINAL CUMULATIVE EDGES

$$(\text{year}, \text{day}), (\text{month}, \text{day})$$

---

---

**Conquer Hypothesis Agent for variables Region/Area**

SPATIAL HIERARCHY EDGES

- region → area
- area → region

---

**Conquer Critic Agent for variables Region/Area**

EVALUATION OF SPATIAL HIERARCHY EDGES

1. EVALUATION OF PROPOSED RELATIONSHIPS

The dataset defines a spatial hierarchy where area is typically nested within region. In a generative sense (e.g., administrative structure or modeling of where events occur), region constrains the set of possible areas. This supports (region, area).

The reverse edge (area, region) implies a causal effect of the specific locality on the broader region, which does not reflect the hierarchical causation. It more closely represents a deterministic data derivation (region can be inferred from area), not a causal influence.

2. ASSESSMENT OF ANTI-CAUSAL DIRECTIONS

The edge (area, region) is anti-causal under intervention semantics. Intervening on area (e.g., selecting a different locality within the same region) does not change the region; however, intervening on region (e.g., focusing on a different country) changes the set of permissible areas. Thus, (area, region) is an effect-to-cause edge and should be removed from the causal graph.

3. CONTEXTS AND COUNTERFACTUALS

In practice, region can be algorithmically derived from area due to hierarchical coding. This yields a functional dependency that may appear as (area, region) in a data-processing DAG. However, causally, the broader spatial context constrains the finer one; therefore, the correct causal direction is (region, area) across typical earthquake datasets.

**Counterfactual examples:**

- If we change region (e.g., from Country A to Country B), the distribution of areas changes (available areas differ).
- If we change area within the same region, region remains the same; choosing an area outside the current region effectively implies changing the region first.

Hence, the causal arrow from region to area is robust.

4. METHODOLOGICAL NOTE

No additional human input required beyond the dataset's provided hierarchy; the conclusion follows from the spatial nesting and intervention logic.

FINAL CUMULATIVE EDGES

$$(\text{region}, \text{area})$$

---

---

**Conquer Hypothesis Agent for Partition Richter/Deaths**

### ANALYSIS OF MAGNITUDE–DEATH RELATIONSHIP

1. CONTEXT UNDERSTANDING

- **richter:** Represents earthquake magnitude on a logarithmic scale, which determines ground-motion amplitude and released energy. Higher magnitude tends to cause stronger shaking and more potential damage.
- **deaths:** Represents recorded fatalities due to each event. Fatalities occur downstream of shaking and damage, and are modulated by many contextual factors (population density, building codes, proximity to epicenter, depth, secondary hazards like tsunamis/landslides, time of day, and reporting practices).

2. ROOT NODES AND RELATIONSHIPS

Among the provided variables, `richter` is a root cause in the physical sense: the magnitude of the event is generated by tectonic processes and is not influenced by the number of deaths.
`deaths` is downstream of `richter`. The causal pathway is:

$$\texttt{richter} \rightarrow \text{shaking intensity/damage} \rightarrow \text{casualties}$$

(Note: Many mediators/moderators are not in the provided variable list).

3. PRELIMINARY EDGES

Potential edge: (`richter`, `deaths`). Larger magnitudes generally increase the risk and extent of damage, increasing fatalities.

4. REFLECTION AND IMPROVEMENT

- **Reverse causation** (`deaths` → `richter`) is not plausible because fatalities do not determine physical magnitude.
- **Confounding** exists (e.g., population density, building standards, depth), but those are not provided variables; the best graph over the two variables is `richter` → `deaths`.
- **Measurement/reporting biases** can affect `deaths` but do not causally impact magnitude; they would be separate variables if present.

FINAL EDGE DETERMINATION

$$(\texttt{richter}, \texttt{deaths})$$

---

**Conquer Critic Agent for Partition Richter/Deaths**

### CAUSAL ANALYSIS OF RICHTER → DEATHS

STEP 1: EVALUATE PROPOSED RELATIONSHIP

**Edge (`richter`, `deaths`):** Plausible and consistent with seismological understanding. Earthquake magnitude reflects energy release and correlates with expected shaking and potential damage. Higher magnitude tends to increase risk of fatalities, all else equal.

STEP 2: ANTI-CAUSAL DIRECTION CHECK

No evidence that `deaths` cause magnitude. Intervening on `deaths` (e.g., improving emergency response) would not change the magnitude of the earthquake; intervening on magnitude (hypothetically reducing energy release) would affect expected fatalities. Therefore, the proposed direction (`richter` → `deaths`) is not anti-causal.

STEP 3: CONTEXT DEPENDENCE AND COUNTERFACTUALS

The relationship is not deterministic and holds only under certain contexts due to strong effect modification:

- **Population exposure and vulnerability:** Dense urban areas with poor building standards can have high deaths even for moderate magnitudes; remote areas may have low deaths for large magnitudes.
- **Depth and proximity:** Shallow earthquakes near population centers produce stronger ground shaking at the surface; deeper events can have high magnitude but lower surface intensity.
- **Secondary hazards:** Tsunamis, landslides, fires, and aftershocks can substantially increase fatalities beyond what magnitude alone would predict.
- **Time of day and preparedness:** Nighttime events or inadequate early warning increase risk.
- **Reporting practices:** Death counts can be underreported or revised; magnitude estimates can be updated post-event, but these do not reverse causality.

**Counterfactuals:**

- Holding magnitude fixed and improving building codes or reducing exposure would reduce deaths.
- Increasing magnitude while holding other conditions constant would generally increase expected deaths.
- Changing deaths does not change the realized magnitude.

STEP 4: ADDITIONAL INFORMATION AND EVIDENCE

Within seismology, magnitude is an exogenous geophysical variable relative to human impact variables; common "confounders" of `deaths` (e.g., population density, building quality, secondary hazards) do not cause magnitude. They primarily act as moderators or mediators along the damage pathway. Thus, while the edge is valid, it is incomplete without adding other parent variables to `deaths`.

FINAL CUMULATIVE EDGES

$$(\text{richter}, \text{deaths})$$

---

---

**Conquer Hypothesis Agent for Partition Year/Month/Day/Region/Area/Richter/Deaths (Part 1 of 2)**

## CAUSAL GRAPH CONSTRUCTION AND EDGE LIST

### STEP 1: CONTEXT UNDERSTANDING

The variables describe individual earthquake events with their occurrence date (`year`, `month`, `day`), location (`region`, `area`), physical intensity (`richter` magnitude), and human impact (`deaths`).

- **Causally:** Magnitude drives shaking and potential harm; spatial context (`region`, `area`) modulates exposure and vulnerability; temporal context (`year`, `month`, `day`) affects vulnerability, exposure, and reporting.
- **Non-physical mechanisms:** Catalog completeness and measurement practices change over time (`year` → `richter`), seasonal/tidal stresses can weakly modulate earthquake timing (`month`/`day` → `richter`), and reporting of deaths evolves over time (`year` → `deaths`).

### STEP 2: ROOT NODES AND HIERARCHICAL RELATIONS

- **Root-like exogenous drivers:** `year`, `month`, `day` (calendar), `region` (broad location).
- **Hierarchies and nesting:**
  - `region` determines `area` (`area` is nested within `region`).
  - `year` determines calendar structure (`year` → `month`; `month` → `day`; and `year` → `day` due to leap-year effects).
- **Physical context:** `region` and `area` influence typical magnitude distributions due to tectonic setting (`region`/`area` → `richter`).
- **Impact:**
  - `richter` → `deaths` is a primary physical causal link.
  - `region`/`area` → `deaths` via population density, building codes, terrain, secondary hazards (landslides, tsunamis).
  - `year`/`month`/`day` → `deaths` via preparedness, exposure patterns, weather, tourism seasons, work/school schedules, and reporting differences.

### STEP 3: PRELIMINARY LIST OF POTENTIAL EDGES

*(Intentionally not conservative)*

- `region` → `area` (Spatial hierarchy).
- `region` → `richter` (Tectonic setting influences magnitude distribution).
- `area` → `richter` (Local fault characteristics).
- `richter` → `deaths` (Physical impact).
- `region` → `deaths` (Vulnerability, exposure, secondary hazards).
- `area` → `deaths` (Finer-grained vulnerability, exposure).
- `year` → `deaths` (Preparedness, building codes, reporting).
- `month` → `deaths` (Seasonal exposure, weather, tourism).
- `day` → `deaths` (Day-of-week/time-related exposure; reporting logistics; tidal/secondary hazard timing).

---

**Conquer Hypothesis Agent for Partition Year/Month/Day/Region/Area/Richter/Deaths(Part 2 of 2)**

- `year` → `richter` (Catalog completeness/measurement practices over time).
- `month` → `richter` (Seasonal hydrological loading; weak but plausible).
- `day` → `richter` (Earth tides/timing effects; weak but plausible).
- `year` → `month` (Calendar hierarchy).
- `month` → `day` (Calendar hierarchy).
- `year` → `day` (Leap-year structure).

STEP 4: REFLECTION AND IMPROVEMENTS

The physical causation of earthquake magnitude is not driven by calendar time, but small seasonal/tidal modulations and catalog/reporting biases justify including `year/month/day` → `richter` as potential edges. Spatial context strongly affects both magnitude distributions and human impact, so `region/area` links to both `richter` and `deaths` are retained. Calendar hierarchy edges are included to reflect structured temporal variables which can influence downstream variables.

The final graph emphasizes direct physical causation (`richter` → `deaths`) and multiple contextual mediators/modulators (`region/area`, `year/month/day`).

FINAL ANSWER: CUMULATIVE EDGES

$$(\text{region}, \text{area}), (\text{region}, \text{richter}), (\text{area}, \text{richter}),$$
$$(\text{richter}, \text{deaths}), (\text{region}, \text{deaths}), (\text{area}, \text{deaths}),$$
$$(\text{year}, \text{deaths}), (\text{month}, \text{deaths}), (\text{day}, \text{deaths}),$$
$$(\text{year}, \text{richter}), (\text{month}, \text{richter}), (\text{day}, \text{richter}),$$
$$(\text{year}, \text{month}), (\text{month}, \text{day}), (\text{year}, \text{day})$$

---

---

**Conquer Critic Agent for Partition Year/Month/Day/Region/Area/Richter/Deaths (Part 1 of 2)**

## CRITICAL EVALUATION AND EDGE REVISION

### STEP 1: EVALUATION OF PROPOSED RELATIONSHIPS

**Hierarchy** (`region`, `area`) As a data hierarchy, `area` is nested within `region`. Deterministically, the `area` label implies its `region`. The direction (`region` → `area`) is questionable because changing the region label does not causally determine the specific area; rather, the area determines which region it belongs to. **Recommendation:** Reverse to (`area`, `region`).

**Location** → **Magnitude** (`region`, `richter`) **and** (`area`, `richter`)**:** Location (tectonic setting, fault characteristics) influences the distribution of magnitudes and the likelihood of large events. These edges are plausible as structural causes of magnitude for a given event.

**Magnitude** → **Deaths** (`richter`, `deaths`)**:** This is a primary physical driver of potential fatalities; higher magnitude generally increases shaking severity, which increases risk of casualties, all else equal. **Valid.**

**Context** → **Deaths** (`region`, `deaths`) **and** (`area`, `deaths`)**:** Contextual mediators like population density, building codes, terrain, and secondary hazards vary by location and modulate the translation of shaking into fatalities. **Valid.**

**Time** → **Deaths** (`year`, `deaths`), (`month`, `deaths`), (`day`, `deaths`)**:** Time can influence exposure and vulnerability (e.g., secular improvements in building standards over years; seasonal tourism; weather conditions that affect landslides/tsunami impacts). These edges are plausible as temporal mediators/proxies.

**Time** → **Magnitude** (`year`, `richter`), (`month`, `richter`), (`day`, `richter`)**:** Calendar time generally does not causally drive earthquake magnitude. While some weak seasonal signals may exist due to hydrological loading or anthropogenic activities, for natural tectonic earthquakes the causal direction is wrong in most contexts. These edges are likely anti-causal and should be **removed**.

**Calendar Structure** (`year`, `month`), (`month`, `day`), (`year`, `day`)**:** These encode calendar structure. As deterministic relationships of an event timestamp, the directions are acceptable for a data-generating graph (`year` determines valid months; `month` and `year` together determine valid days including leap years).

### STEP 2: ANTI-CAUSAL DIRECTION CHECKS

- (`region`, `area`): More defensible as (`area`, `region`) because `area` implies `region`. Intervening on `area` (choosing a locality) changes `region`; intervening on `region` does not uniquely determine `area`.

- **Time** → **Richter:** Anti-causal for tectonic events. Intervening on calendar date does not change the physical magnitude of a specific earthquake.

**Conquer Critic Agent for Partition Year/Month/Day/Region/Area/Richter/Deaths (Part 2 of 2)**

STEP 3: CONTEXTUAL VALIDITY AND COUNTERFACTUALS

- **region/area → richter:** Holds in the sense of geological control on possible magnitudes and frequency distributions. Counterfactual changes in location (different faults/tectonic regimes) change magnitude distribution.

- **richter → deaths:** Generally holds but can be overridden by counterfactuals with strong mitigation (e.g., high magnitude with few deaths in sparsely populated areas or where buildings are resilient).

- **region/area → deaths:** Context-specific; strong in regions with high exposure/vulnerability; weaker where resilience is high.

- **year → deaths:** Context-specific; secular improvements or degradations over time affect vulnerability and reporting. Counterfactual changes in year (as a proxy for era) can change expected deaths via evolving codes and emergency response.

- **month/day → deaths:** Seasonal exposure and secondary hazards can modulate fatalities (e.g., monsoon-related landslides, tourist seasons). Context-dependent.

STEP 4: ADDITIONAL CONSIDERATIONS

Important missing drivers of deaths include hypocentral depth, distance to populated areas, building code enforcement, time of day (occupancy), and secondary hazards (tsunami/landslide triggers). These confound richter → deaths and region/area → deaths pathways. Reporting completeness may also vary by region and year, confounding deaths.

EDGE LIST EVOLUTION

**Iteration 1 (Original Hypothesis):**

(region, area), (region, richter), (area, richter), (richter, deaths),
(region, deaths), (area, deaths), (year, deaths), (month, deaths),
(day, deaths), (year, richter), (month, richter), (day, richter),
(year, month), (month, day), (year, day)

**Iteration 2 (Revised Edges):** *Reversed hierarchical direction; removed anti-causal time → magnitude edges; retained contextual and temporal mediators.*

(area, region), (region, richter), (area, richter), (richter, deaths),
(region, deaths), (area, deaths), (year, deaths), (month, deaths),
(day, deaths), (year, month), (month, day), (year, day)

---

**Conquer Hypothesis Agent for Partition Year/Deaths**

## CAUSAL ANALYSIS: YEAR VS. DEATHS

### 1. CONTEXT UNDERSTANDING

- **year:** Represents the calendar time of the earthquake event. At a causal level, `year` is a proxy for secular changes such as urbanization, exposure, building codes, enforcement, infrastructure resilience, emergency response capacity, and reporting completeness. These time-varying factors can influence the number of recorded deaths per event.

- **deaths:** Represents fatalities attributed to the specific earthquake event. It is affected by many unobserved event-specific and context-specific factors (magnitude, depth, proximity to population, building vulnerability, secondary hazards, response quality), many of which evolve over time and are partially captured via `year`.

### 2. ROOT NODES AND RELATIONSHIPS

- **year** is a root node in this two-variable system: it is exogenous and not caused by `deaths`.

- **year** can influence `deaths` via:
    - True vulnerability/exposure changes over time (population growth, urbanization patterns).
    - Improvements in building codes and enforcement reducing fatalities.
    - Changes in emergency response capacity and medical care reducing fatalities.
    - Changes in reporting completeness potentially increasing recorded deaths even if true deaths are stable.

- **deaths** does not influence `year`. The calendar date is determined independently of the outcome. While reporting practices might make certain events more likely to be recorded, that does not causally change the `year` variable of an event.

### 3. PRELIMINARY LIST OF POTENTIAL EDGES

*(Being expansive)*

- (year, deaths): Plausible and expected.
- (deaths, year): Considered but likely implausible because outcomes do not set calendar time.

### 4. REFLECTION AND IMPROVEMENT

The only defensible causal direction between these two variables is year $\rightarrow$ deaths. Year functions as a surrogate for multiple evolving causal mechanisms that affect fatalities and for reporting-era effects that change recorded deaths. There is no credible mechanism whereby deaths would cause the year of occurrence.

### FINAL EDGE DETERMINATION

$$(\text{year}, \text{deaths})$$

---

**HITL Call**

In your earthquake dataset, do you expect reporting completeness and exposure (population/building density) to have changed over time, making year a proxy for such secular trends? Or is the dataset already harmonized to control for these factors?

**User Response**

This dataset just contains raw measurements and recorded data. No preprocessing was made.

**Conquer Critic Agent for Partition Year/Deaths (Part 1 of 2)**

EVALUATION OF THE RELATIONSHIP: YEAR → DEATHS

STEP 1: EVALUATE PROPOSED RELATIONSHIP

**Plausibility** year itself does not directly cause fatalities, but it is a proxy for secular changes that do influence earthquake mortality. Over time, urbanization (exposure), building codes and construction quality (vulnerability), emergency response capacity, and reporting completeness tend to change. Because the dataset is raw (no harmonization), year can capture these trends, making an association between year and recorded deaths plausible.

**Mechanisms**
  • **Exposure:** Urbanization increases population density near fault zones, potentially increasing deaths for similar magnitudes.
  • **Vulnerability and resilience:** Improved building standards and response capacity can reduce deaths over time in some regions.
  • **Reporting completeness:** Later years often have more complete death counts; earlier years may underreport, inflating an apparent temporal trend.

**Conclusion** Modeling year → deaths is reasonable as a shorthand for unobserved time-varying causes and measurement processes.

STEP 2: ASSESS ANTI-CAUSAL DIRECTION

The proposed edge (year, deaths) is not in the anti-causal direction. Intervening on deaths would not change the calendar year, while interventions on time-varying factors aligned with year (e.g., building codes) can change deaths.
*Note:* year itself is not manipulable, but it indexes manipulable processes. So year → deaths is a proxy relationship rather than a direct causal mechanism.

STEP 3: CONTEXT DEPENDENCE AND COUNTERFACTUALS

  • **Context dependence:** The relationship may vary by region and era. In some contexts, modernization reduces deaths; in others, rapid urbanization without adequate codes increases deaths. Thus, the sign and magnitude of the year effect are not uniform.
  • **Narrow windows:** Within short time spans or within homogeneous regions with stable exposure/vulnerability, the year → deaths association may be weak or negligible.
  • **Counterfactuals:** Holding magnitude, region, and exposure constant, deaths could change over calendar time due to building code improvements; conversely, if exposure increases faster than safety measures, deaths could rise with year.

## Conquer Critic Agent for Partition Year/Deaths (Part 2 of 2)

STEP 4: ADDITIONAL EVIDENCE AND CONSIDERATIONS

Since the dataset is raw with no harmonization, reporting completeness and exposure likely change with `year`. This supports treating `year` as a proxy cause of recorded deaths via measurement error and secular trends.

**Potential confounding and omitted variables:**

- **Magnitude (`richter`):** Strongly affects deaths.
- **Region/Area:** Determine exposure and vulnerability.
- **Event depth, distance, secondary hazards:** Further mediate deaths (not present in dataset).

Without controlling for these, the estimated `year` → `deaths` effect may be biased.

FINAL CONCLUSION AND EDGE LIST

The edge (`year`, `deaths`) is directionally reasonable as a proxy for time-varying causes and reporting changes, but it is context-dependent and should be interpreted as capturing secular trends rather than a direct physical cause. No anti-causal issues are identified with the proposed direction.

$$(\texttt{year}, \texttt{deaths})$$

## Conquer Hypothesis Agent for Partition Region/Area/Months/Deaths

SPATIAL AND TEMPORAL CONTEXT EDGES

- `region` → `area`
- `region` → `deaths`
- `area` → `deaths`
- `month` → `deaths`

## HITL Call

In your experience with earthquake impacts, does seasonal timing (month) significantly modulate fatalities via secondary hazards (e.g., monsoon-triggered landslides, winter cold, tourist seasons) across regions, or is its effect typically negligible compared to magnitude and proximity?

## User Response

I don't know.

## HITL Call

Beyond local area characteristics (population density, building standards), do broader regional factors (e.g., tsunami exposure, monsoon regimes, disaster preparedness) exert a direct influence on earthquake fatalities, or are they mostly mediated through the specific area-level conditions?

**User Response**

They likely have an influence, as for example certain areas might just be more densely populated.

---

**Conquer Critic Agent for Partition Region/Area/Month/Deaths (Part 1 of 2)**

## EVALUATION OF SPATIAL AND TEMPORAL EDGES

### STEP 1: EVALUATE PROPOSED RELATIONSHIPS

**Hierarchy (`region`, `area`) Valid.** `area` is hierarchically nested within `region`; regional context determines which areas exist and their hazard environment. Intervening on `area` (choosing a different locality) does not retroactively change the higher-level region definition; conversely, changing `region` necessarily changes the set of possible areas.

**Region Effect (`region`, `deaths`) Plausible but largely mediated by `area`.** Regional characteristics (tsunami exposure, monsoon regimes, governance and preparedness) can influence fatalities, but in most practical analyses, these act through local area-level conditions (population density, construction quality, terrain, emergency response capabilities). Without additional variables, including a direct `region` → `deaths` edge risks conflating mediation with a direct effect.

**Local Effect (`area`, `deaths`) Valid.** Local vulnerability and exposure strongly modulate fatalities (building standards, terrain-related secondary hazards, proximity to population).

**Seasonal Effect (`month`, `deaths`) Context-dependent.** Seasonal timing can affect fatalities via secondary hazards (e.g., rainfall-triggered landslides) and exposure (tourist season peaks), but its effect magnitude varies by region and hazard profile. It is exogenous and can causally influence conditions that modulate death tolls.

### STEP 2: ANTI-CAUSAL DIRECTION ASSESSMENT

None of the proposed edges are anti-causal in orientation. Intervening on `deaths` will not change `region`, `area`, or `month`. `region` → `area` is correctly oriented (not `area` → `region`). `month` → `deaths` is correctly oriented given `month` is exogenous calendar timing.

### STEP 3: DO RELATIONSHIPS ALWAYS HOLD?

- (`region`, `area`): Holds universally as a definitional/hierarchical relation.
- (`area`, `deaths`): Holds broadly; there are counterfactuals where even high-vulnerability areas see low deaths (e.g., deep, offshore events), but within this variable set, `area` remains a causal determinant of deaths.
- (`month`, `deaths`): Holds in contexts with strong seasonal secondary hazards or exposure cycles; may be negligible in tectonically dominated, arid, or less seasonally variable regions.
- (`region`, `deaths`): Often mediated by `area`; may hold directly in contexts where region-level policies or hazard exposures are not captured by the `area` variable (e.g., coastal tsunami belts spanning multiple areas, uniform regional building codes). Otherwise, it should be treated as indirect via `area`.

---

**Conquer Critic Agent for Partition Region/Area/Month/Deaths (Part 2 of 2)**

STEP 4: TOOL-SUPPORTED EVIDENCE AND RECOMMENDATIONS

Human input indicates uncertainty about the magnitude of the effect of `month` and suggests broader regional influences are often expressed via area-level characteristics, supporting mediation of (`region`, `deaths`).

**Recommended modifications:**

- Retain (`region`, `area`), (`area`, `deaths`), (`month`, `deaths`).
- Treat (`region`, `deaths`) as primarily mediated by `area`; include it only if analysis intends to capture region-level effects not encoded in `area` (e.g., preparedness or hazard regime differences spanning areas). Otherwise, remove to avoid over-specification and double-counting.

FINAL ACCEPTED EDGES

$$(\text{region}, \text{area}), (\text{area}, \text{deaths}), (\text{month}, \text{deaths})$$

---

## H.4 MERGE PHASE

In the final merge phase, the merge agents $\mathcal{M}_{hyp}$ and $\mathcal{M}_{critic}$ respectively propose and refine the causal connections between partitions. In the partition tree, this procedure consists in collapsing the partition tree bottom-up by merging the leaves into their respective parent. In case of iterated partitioning during the Divide phase, also the Merge phase will be iterated in the same way. Once the root-node of the partition tree is reached, CAST finishes.

---

**Merge Hypothesis Agent**

DIRECTED EDGES

- (year, deaths)
- (month, deaths)
- (day, deaths)
- (region, deaths)
- (area, deaths)
- (richter, deaths)
- (region, richter)
- (area, richter)
- (year, richter)

---

> **Merge Critic Agent (Part 1 of 2)**
>
> ## ANALYSIS OF CAUSAL RELATIONSHIPS
>
> ### RELATIONSHIP EVALUATION
>
> **Time → Deaths**
> - (`year`, `deaths`): Plausible. Secular changes (building codes, emergency response, urbanization, reporting) can change fatality risk and documentation over time.
> - (`month`, `deaths`): Plausible. Seasonal conditions (e.g., monsoon, winter storms) and exposure patterns can modulate secondary hazards and vulnerability.
> - (`day`, `deaths`): Weak/context-dependent. Day-of-month alone is a crude proxy; without day-of-week/holiday information, its causal influence is limited, though specific calendar effects could exist in some contexts.
>
> **Location → Deaths** (`region`, `deaths`) **and** (`area`, `deaths`): Plausible. Spatial context (population density, construction quality, site effects, terrain, tsunami/landslide exposure) modulates fatalities given shaking.
>
> **Magnitude → Deaths** (`richter`, `deaths`): Strong and well-supported. Larger magnitude generally increases shaking potential and risk of casualties, though the effect is moderated by location and context.
>
> **Location → Magnitude** (`region`, `richter`) **and** (`area`, `richter`): Not appropriate as direct causal edges for event-level data. Region/area are descriptors of where the event occurred; the underlying tectonic setting influences both where earthquakes occur and the distribution of magnitudes, but the recorded "region/area" label does not cause an event's magnitude. A more accurate structure would include an unobserved tectonic/structural variable causing both location and magnitude, or treat region/area as stratifying variables rather than direct causes of magnitude.

---

**Merge Critic Agent (Part 2 of 2)**

**Time → Magnitude (`year`, `richter`):** Generally not causal. Calendar year does not cause the physical magnitude of an earthquake. Apparent temporal trends in recorded magnitudes can reflect reporting/instrumentation changes (selection/measurement effects), but the true magnitude of an event is not determined by year. Modeling this as a direct causal edge risks conflating catalog biases with physical causation.

ANTI-CAUSAL DIRECTION ASSESSMENT

Edges into `richter` from `region`, `area`, and `year` are in an anti-causal or non-causal direction for event-level causality. Intervening on these labels would not change the physical magnitude of a given event.

CONTEXT-DEPENDENCE

- (`year`, `deaths`) and (`month`, `deaths`) hold in many contexts but are mediated by evolving vulnerability, exposure, and seasonal hazards.
- (`day`, `deaths`) is especially context-dependent and likely weak unless coupled with additional information (e.g., day-of-week, holidays).
- `region`/`area` → `deaths` and `richter` → `deaths` are broadly applicable, though moderated by unobserved factors (e.g., depth, distance to population centers).

REVISED EDGES

- **Keep:** (`year`, `deaths`), (`month`, `deaths`), (`region`, `deaths`), (`area`, `deaths`), (`richter`, `deaths`).
- **Optionally keep:** (`day`, `deaths`) with the caveat that it is weak/context-dependent.
- **Remove:** (`region`, `richter`), (`area`, `richter`), (`year`, `richter`).

FINAL RECOMMENDED EDGE SET

- (`year`, `deaths`)
- (`month`, `deaths`)
- (`day`, `deaths`)
- (`region`, `deaths`)
- (`area`, `deaths`)
- (`richter`, `deaths`)

---

H.5 FINAL RESULT

After combining the all edges discovered during the conquer and merge phase, the final causal graph is reported in 7.

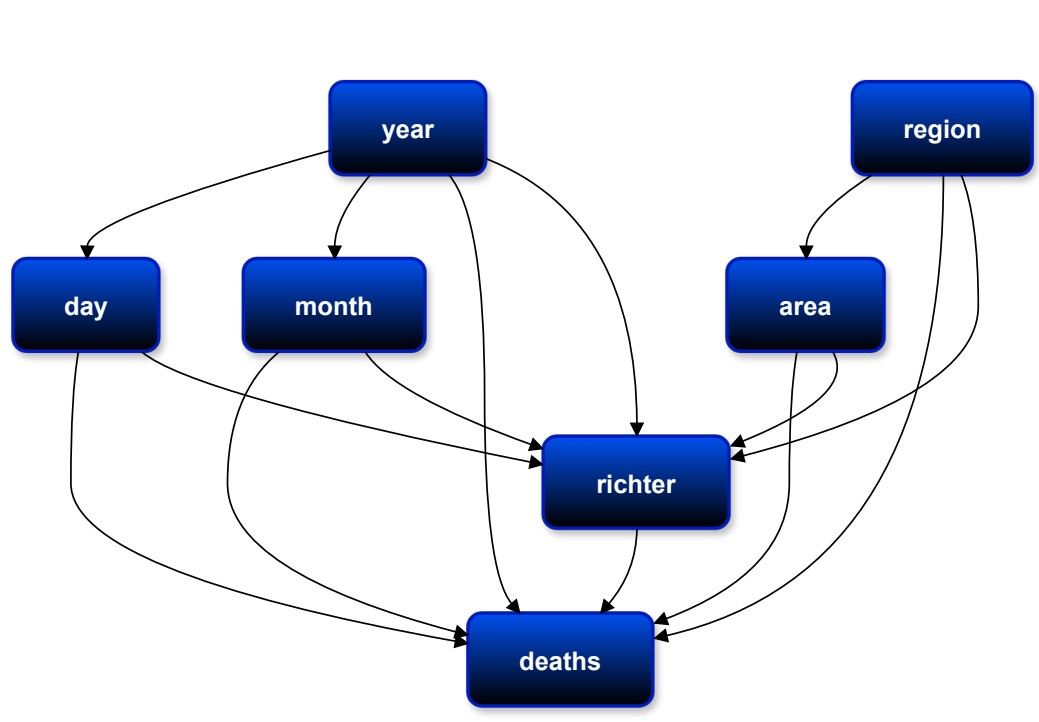

Figure 7: Causal Graph Discovered by CAST for the Earthquakes dataset.

