# OpenReview forum: "CausalSteward: An Agentic Divide-Conquer-Combine Copilot for Causal Discovery"
_ICLR.cc/2026/Conference — Submitted to ICLR 2026_

### Official Review · Reviewer_8q3g · 2025-10-22

**Soundness:** 2
**Presentation:** 3
**Contribution:** 3
**Rating:** 4
**Confidence:** 3

**Summary:**

LLMs have recently been applied to providing background knowledge to causal discovery algorithms; however LLMs struggle in graphs with many nodes. This submission proposes a new method that employs divide-and-conquer to partition the graph into smaller graphs, that LLMs can then infer. These are combined into a larger graph.

**Strengths:**

- Original approach, as I have never seen divide-and-conquer in this context, while it does naturally fit into helping with large graphs.

- Theoretical properties; whose proofs look correct.

- Extensive empirical simulations

**Weaknesses:**

There are some glitches in experiments:
- LLM-BFS and Pairwise only seem evaluated on one LLM each, and a different one, making it hard to rule out an effet from the LLM alone in these baselines.
- "Therefore, we confirm (Q3)." (l.429). What do you confirm? (Q3) is an open-ended question, not a statement.
- Baselines are not evaluated on CausalChambers, and I actually do not see any results for Earthquake.
- There is no ablation study for the Explainer.
- ""Regarding scalability, the decrease in performance from CausalMan Small to Medium is only sub-linear, whereas data-driven baselines degrade by a factor of 4-6, while C A ST decreases sub-linearly by a factor of ≈ 2" (l.416-418) : for which methods and metrics? There is little change for BOSS on F1, and a sub-2 decrease in performance for XGES on SHD.
- CAST HITL+RAG is not consistently better than HITL or RAG alone: only 2 out of 6 instances on F1 in Table 1

**Questions:**

Can you answer/address the above?

In addition, which humans are interacting with the model? If these are authors themselves, how do you ensure that comparisons with baselines fair?

---

> ### Author Response · Authors · 2025-11-23
> **Answers and Clarifications**
>
> Dear Reviewer 8q3g,
>
> Thanks for reviewing our work and for the feedback.
>
> We updated our manuscript by implementing your detailed feedback. Your color in the manuscript is $\color{red}{\text{red}}$. Feedback shared by more than one reviewer in $\color{violet}{\text{violet}}$.
>
> ## Answers to Questions and Weaknesses
>
> - **[Evaluation of LLM-BFS and Pairwise]:** We tested using the official implementation of both methods. With said implementation, we couldn't run LLM-BFS on Qwen3-14B due to limitations in the context-window, therefore we opted for o3-mini. For Pairwise, we did choose Qwen3 instead since it is better performing than o3-mini.
> - **[Precisation on Q3]:** We recognized that Q3 was phrased in an open-ended manner. We rephrased it in a clearer way in the new version of the manuscript, where Q3 now is "Do computational requirements scale tractably with the size of the graph?", to which our answer is yes.
> - **[Baselines for CausalChambers]:** We hereby provide the performance for baselines on CausalChambers. We used 10.000 samples. $F_1$ and Precision are (in most cases) an order of magnitude better than baselines, from which we deduce that CAST has superior performance in skeleton recovery. This means that CAST correctly finds the majority of true causal links. The seemingly "worse" Structural Hamming Distance (SHD) is not a failure but more a consequence of edge orientation errors. Since False Positives and False Negatives are low (due to high F1), the high SHD is almost entirely driven by the penalty that SHD applies to incorrectly oriented edges, highlighting that directional inference needs refinement.
>
>     | Method | $F_1 \uparrow$ | Precision$\uparrow$ | SHD$\downarrow$ |
>     |-------|-------|-------|-------|
>     | XGES | 0.053(1e-4) | 0.091(1e-4)  | 81(0.0) |
>     | FCI | 0.059(0.011) | 0.054(0.009)  | 84.4(3.6) |
>     | BOSS | 0.091(1e-4) | 0.081(1e-4) | 82(0.0) |
>     | GranDAG | 0.032(1e-4) | 0.233(0.037)  | **59.4(0.9)** |
>     | CAST  | **0.411(0.086)** | **0.315(0.059)**  | 96.6(11.81) |
>
> - **[Results for Earthquake]:** Results for earthquake are present in Appendix B. The dataset is real-world and, as it happens in most real-world applications of causal discovery, the ground truth graph is not available, therefore metrics were not evaluated. The dataset is used as a real-world case-study in [1], and we similarly used it to illustrate how the interaction with CausalSteward flows in a relatively simple dataset.
> - **[Ablation for Explainer]:** Normally, the explainer expands the (high-level) user description and outputs a description for every single variable, which we store and use during the next steps. In this ablation, we skipped that phase, therefore all subsequent agents performed their operation with only a the brief initial description made by the user, without any expansion of that knowledge made by the explainer agents. Below our results. We observe that the base version of CAST is better on CausalMan Small, and shows a more complex outlook in CausalMan Medium. Indeed, without explainer, the prior knowledge used by LLMs is less organised and they become more prone to ambiguities in the variable naming, therefore the agents act hyper-conservatively in adding new edges. This leads to a more sparse graph which brings down recall (therefore $F_1$) and minimally improves SHD w.r.t. the base version of CAST.
>
>     - For CausalMan Small:
>         | Variant | $F_1 \uparrow$ | Precision$\uparrow$ | SHD$\downarrow$ |
>         |-------|-------|-------|-------|
>         | No Explainer  | 0.239(0.107) | 0.453(0.115) | 120.20(16.94) |
>         | No Divide | 0.34(0.06) |0.34(0.08) | 199.0(20.3) |
>         | CAST  | **0.42(0.08)** | **0.72(0.17)**   | **103.0(10.1)** |
>
>     - For CausalMan Medium:
>         | Variant | $F_1 \uparrow$ | Precision$\uparrow$ | SHD$\downarrow$ |
>         |-------|-------|-------|-------|
>         | No Explainer  | 0.154(0.044) | 0.589(0.158) | **540.00(19.89)** |
>         | No Critic | **0.31(0.02)**| 0.56(0.13)| 588.0(43.2) |
>         | No Divide | - |- | -  |
>         | CAST  | 0.27(0.05)| **0.66(0.11)** | 558.0(21.9)|
> - **[Performance scaling]:** For F1: XGES, GranDAG, and CausalCopilot show a severe decrease. BOSS is more stable, and we added this remark to the manuscript.
> - **[HITL+RAG Performance]:** We do not claim that the HITL+RAG is strictly better in performance. In large-scale applications, a purely HITL implementation results in a large number of user interactions, which hinders scalability. Therefore, we presented our HITL+RAG implementation as an intermediate solution between the purely automatic RAG and the HITL version.
>
> We hope that this message gave clarity to our work. If additional questions, feedback or clarifications are needed, you are welcome continue this discussion.
>
> The authors
>
> ---
>
> ## References
>
> - [1] "Causal-Copilot: An Autonomous Causal Analysis Agent", Wang et al., 2025

---

> > ### Comment · Reviewer_8q3g · 2025-11-25
> >
> > Thanks for your helpful rebuttal! Here are remaining points:
> >
> > > For Pairwise, we did choose Qwen3 instead since it is better performing than o3-mini.
> >
> > Is there a proof for this? Can you eg also add Pairwise on o3-mini?
> >
> > > For F1: XGES, GranDAG, and CausalCopilot show a severe decrease. BOSS is more stable, and we added this remark to the manuscript.
> >
> > I understand, so by "sub-2" or "several times" decrease in performance, you referred to F1? I do not see the remark on BOSS being stabler in the edited manuscript.
> >
> > ****
> >
> > It also looks like you did not address this remark: **In addition, which humans are interacting with the model? If these are authors themselves, how do you ensure that comparisons with baselines fair?**

---

> > > ### Author Response · Authors · 2025-11-25
> > > **Additional points**
> > >
> > > Dear reviewer 8q3g,
> > >
> > > Thanks for engaging with our rebuttal! We clarify your points below:
> > >
> > > - **[Pairwise querying using o3-mini]** Yes and we provide here the new results. Below, we compare LLM-Paiwise [1] with o3-mini and Qwen3-14B. On the o3-mini LLM, we kept the default $temperature$ parameter (which can either be set to 0 or 1, and is 0 by default). With $temperature=1$ we measured an exaggerate variability and overall decrease in all metrics, therefore we kept it to the default value $=0$. As a result, LLM-Pairwise with o3-mini showed very low std in every metric (similar also for LLM-BFS). We confirm here that, for LLM-Pairwise, Qwen3-14B consistently yields superior performance compared to o3-mini. Results on CausalMan Small.
> > >     | LLM | $F_1 \uparrow$ | Precision$\uparrow$ | SHD$\downarrow$ |
> > >     |-------|-------|-------|-------|
> > >     | Pairwise(o3-mini)  | 0.126(0.004) | 0.074(0.003) | 710(6.2) |
> > >     | Pairwise(Qwen3-14B) | **0.153(0.008)** | **0.093(0.005)** | **583.50(11.1)** |
> > >
> > > - **[BOSS Performance]** We checked and apologize for it. That remark got lost among other corrections. We uploaded a new version of the manuscript that makes it clearer. Now it writes "the $F_1$ decrease in performance from CausalMan Small to Medium is only sub-linear (by a factor of $\approx 2$) for CAST, whereas most data-driven baselines (XGES, GranDAG, CausalCopilot) degrade by several times. BOSS exhibits stronger robustness, but absolute metrics are still inferior with respect to CAST. Overall, results indicate that CAST scales robustly while keeping good performance, affirming **(Q2)**".
> > >
> > > - **[HITL Subjects]** We have access to human experts that helped validating CausalSteward, and those people were credited by including them among the authors. In essence, **the people who did ideate, design and implement CausalSteward are different from the people who participated in its bechmarking/validation.**
> > >
> > > Best regards,
> > >
> > > The Authors
> > >
> > > ---
> > >
> > > ## References
> > >
> > > - [1] "Causal Reasoning and Large Language Models: Opening a New Frontier for Causality", Kıcıman et al., 2024, TMLR

---

> > > > ### Comment · Reviewer_8q3g · 2025-11-28
> > > >
> > > > Many thanks, this really solidifies the paper! Final question: can you elaborate more on who are these people, how they were recruited, .... ?

---

> > > > > ### Author Response · Authors · 2025-11-28
> > > > > **Human Participants**
> > > > >
> > > > > Dear reviewer 8q3g,
> > > > >
> > > > > Thank you for the positive feedback! We are glad the additional details helped.
> > > > >
> > > > > Regarding the participants: To respect the double-blind policy, we cannot name the specific organization, but we can clarify that the participants are domain experts from a major industrial partner. They were recruited based on their specific experience with the business use-cases used in the evaluation.
> > > > >
> > > > > We chose this group because CAST was ideated as an "Expert-in-the-Loop" system. Therefore, the validation happened with people having a deep knowledge on the system rather than crowdsourced workers. We think this experimental setting given an outlook of CAST that is more aligned with real-world performance.
> > > > >
> > > > >
> > > > >
> > > > > Best regards,
> > > > >
> > > > > The Authors

---

### Official Review · Reviewer_C7Er · 2025-10-23

**Soundness:** 3
**Presentation:** 2
**Contribution:** 2
**Rating:** 4
**Confidence:** 4

**Summary:**

This paper proposes CAST for causal discovery, and CAST is the first attempt to aggregate human-in-loop, multi-agent collaborate and divide&conquer mechanisms. Experiments on  medium to large datasets all demonstrate its superiority on scability, and CAST achieves a good balance between F1 score and runtime.

**Strengths:**

1.	The paper provides sufficient theoretical analysis, including proofs for the ideal case and error bounds under realistic conditions.
2.	It’s the first attempt to integrate human-in-loop, multi-agent collaborate and divide&conquer to solve the causal identifiability and high dimensionality data challenges in causal discovery.
3.	Experiments show the superiority of the proposed method.

**Weaknesses:**

1.	The paper’s novelty may be somewhat limited, as its main contribution seems to lie in aggregating existing techniques. It would be helpful if the authors could better clarify the specific methodological innovations beyond this integration.
2.	The paper reports results for CAST without HITL but does not explain how the “human-in-the-loop” setting was conducted. There is no description of real human involvement or feedback collection. This makes the claimed benefit of human–AI collaboration unclear.
3.	The paper does not evaluate the effect of using different causal discovery algorithms within the Conquer phase. It remains unclear whether the observed improvements are due to the LLM hypothesis, the underlying causal discovery method, or their combination. An ablation study with alternative algorithms could clarify the contribution of each component.
4.	There is a minor error in the subsection Ablation for Divide & Conquer and Ablation for Critic Agents, it should refer to Figure 4 rather than Table 4.

**Questions:**

1.	How was the human-in-the-loop setting implemented? Can the authors provide quantitative or qualitative evidence for the claimed benefits of HITL?
2.	Can the authors clarify which aspects of their method constitute a novel contribution beyond the aggregation of existing techniques?
3.	Have the authors evaluated alternative causal discovery algorithms within the Conquer phase? How much of the observed performance improvement is attributable to the LLM hypothesis versus the underlying causal discovery method?
4.	In the Ablation subsections for Divide & Conquer and Critic Agents, the text refers to Table 4, but it should refer to Figure 4.

---

> ### Author Response · Authors · 2025-11-23
> **Answers and Clarifications**
>
> We thank reviewer C7Er for the feedback, and for acknowledging the strengths of our work.
>
> Indeed, in our vision we want to tackle real-world causality with a human-centered approach, which proves to be critical due to known theoretical limitations of causal discovery from observational data.
>
> Corrections in the manuscript implemented from your feedback are implemented in $\color{green}{\text{green}}$. Feedback by more than one reviewer in $\color{violet}{\text{violet}}$. We corrected the typo and inserted the new results in Figure 5.
>
> ## Answers to Questions and Weaknesses
>
> - **[Contribution and Vision of our work]** In our vision, we seek for interactively building causal models. Therefore, we focus our contribution on the interactive HITL approach to causal structure learning, and on how that is used to tackle real-world large-scale causality. Due to known limitations when working only on observational data, those human priors are crucial. Prior work, in comparison to us, incorporates expert knowledge by either requiring structured inputs (e.g., explicit edge-level constraints, templates, or user-defined parametric priors) or assuming access to a lot of precise prior information. In CAST, and also many in real-world settings, experts interact in limited amount and with an high-level/abstract language. Therefore, **in this paper we propose a method to build large-scale causal models using natural language, and all the steps that we adopted serve this task and are needed to keep interaction scalable (nr.Interactions should not grow exponentially with the size of the graph)**. This is done by balancing RAG with HITL calls, while keeping it tractable by using Divide&Conquer. The reason why our method is more than just the combination of its components is due to the challenge of integrating such human-feedback into large-scale causal discovery: Experts cannot be asked thousands of times about causal edges and often talk at an high level. Therefore, all stages of CAST as a whole enable that, and our contribution stands on the collective behavior of the whole system.
>
> - **[Human-In-The-Loop Experiments]** HITL experiments were performed with the help of real domain experts in manufacturing, which have been involved exclusively in the testing phase (not in the design of CAST). We also recognize that this information was not explicitly indicated, therefore we added a remark in the main text, plus a reproducibility statement, and an ethics statement highlighting the opportunities and risks of integrating human feedback into multiagentic system basd on LLMs.
> - **[Human-In-The-Loop Calls]**  Agents call the human expert in the form of a tool-call, therefore the agents decide when to query the human. Clearly some LLMs are better than others at fullfilling this task, as that depends on their training data and on their instruction-following capabilities.
> - **[Alternative Causal Discovery Algorithms]** Yes, we experimented also a version with the gradient--based DAGMA and with the basic FCI without prior knowledge. For DAGMA [1], we did not integrate prior edges as adding edge constraints in the loss, as that might make the training unstable. Therefore, we did plug-in LLM's edges after running the method. Here are our new results:
>
>     - | **Method** | $F_1$$\uparrow$ | Precision $\uparrow$ | $SHD\downarrow$ |
>         |-------|-------|-------|-------|
>         | DAGMA  | 0.1(0.3) | 0.1(0.3)  | 4.5(1.8)  |
>         | DAGMA + Agents | **0.6(0.3)**  | **0.4(0.4)**  | **3.1(2.9)**  |
>         | FCI  | 0  | 0  | 4.6(4.7)  |
>         | FCI+Agents  | 0.3(0.4)   | 0.4(0.5)  | 4.3(5.3) |
>
>     - When we write DAGMA or FCI, we refer to the basic version that does not use any edge given by LLMs. When we write "+Agents" we instead are injecting the LLM's proposed edges.
>     - Since those algorithms are run for every partition, the evaluation for each run is made by evaluating on each partition, and then averaging across them. Then we repeat the runs and average across seeds. The difference in size across partitions is the reason why sometimes the std is higher than the mean value of the metric.
>     - We remark that, by using DAGMA, theoretical guarantees might not hold anymore: DAGMA is not a PAG-learner that aims to learn the Markov Equivalence Class (MEC). Still, we know from recent works [2] that when using certain loss functions there is convergence towards the sparsest DAG in the MEC, so it might be possible that the theory could be readapted.
>
> - **[Typo]** Corrected.
>
> We thank again the reviewer, and hope that this clarified our work and vision. If any additional question or feedback is present, we encourage to continue this discussion.
>
>
> ---
>
> ### References
>
> [1] "DAGMA: Learning DAGs via M-matrices and a Log-Determinant Acyclicity Characterization", Bello et al., NeurIPS 2022
>
> [2] "Markov Equivalence and Consistency in Differentiable Structure Learning", Deng et al.,NeurIPS 2024

---

> > ### Comment · Reviewer_C7Er · 2025-11-26
> >
> > Thank you for the added experiments, they provide convincing empirical support.
> > One remaining point is that the manuscript still does not fully clarify how real human experts interacted with the system in practice. More detail on the exact interaction process would improve transparency.

---

> > > ### Author Response · Authors · 2025-11-27
> > > **New added content**
> > >
> > > Dear reviewer C7Er,
> > >
> > > Thanks for engaging with our rebuttal, and for acknowledging that the new results are convincing!
> > >
> > > To address your point, we think that the most transparent and direct way is to provide full end-to-end interactions of CausalSteward. Therefore:
> > > - **[Earthquakes in Appendix H]** We added one for the Earthquakes dataset (7 variables), which fits more the role of an illustrative example. To insert it into the manuscript, we did polish its layout and remove tags used to parse the LLM's output.
> > > - **[CausalChambers in Supplementary Material]** We also tested on the CausalChambers dataset and exported it as a pdf file. We inserted it in the supplementary material, as keeping it in the paper too would have made the appendix unreasonably long.
> > >
> > > Some interesting key take-aways/observations:
> > > - As it is shown in the HITL calls, it is not necessary to provide explicit/formal causal information of the kind "A implies B" as the interaction happens in natural language, and the extraction of causal information from user feedback is managed by the agents (Supported by [1]).
> > > - Often, agents received high-level, partial, or even "I don't know" responses. This highlights CAST's ability to handle realistic, imperfect user input.
> > > - You may observe occasional repeated queries. This results from the current design choice to keep agents separate without a global shared memory. While this ensures modularity, we acknowledge that adding a shared memory is a straightforward engineering optimization for future work to reduce token usage and enhance scalability further.
> > >
> > > Best regards,
> > >
> > > The Authors
> > >
> > > ---
> > >
> > > ## References
> > >
> > > [1] "Zero-shot Causal Graph Extrapolation from Text via LLMs", Antonucci et al., XAI4Sci Workshop@AAAI24

---

### Official Review · Reviewer_WiZQ · 2025-11-01

**Soundness:** 2
**Presentation:** 1
**Contribution:** 2
**Rating:** 4
**Confidence:** 3

**Summary:**

The paper introduces CausalSteward (CAST), a multi-agent human-in-the-loop (HITL) framework for causal discovery. CAST uses a divide-conquer-combine paradigm that integrates retrieval-augmented generation, human interaction, and data-driven causal discovery algorithms.
The system decomposes high-dimensional variable sets into causally coherent partitions (divide), estimates local causal graphs through LLM-assisted hypothesis and critic agents combined with constraint-based discovery (conquer), and merges these into a global graph (combine).
The authors provide a theoretical analysis demonstrating consistency under ideal causal partitioning (Theorem 1) and decomposing structural errors according to partition and merge inaccuracies (Theorem 2).
They then compare CAST to other baselines with experiments on the CausalMan, Neuropathic-Pain, and CausalChambers benchmarks.

**Strengths:**

- CAST is the first agentic framework combining Divide-and-Conquer, RAG, and human-in-the-loop mechanisms for causal discovery.
- Theorem 2 provides a useful decomposition of SHD in terms of partitioning, merging, and LLM components. The formalization of causal partitioning and its link to theoretical guarantees is valuable.
- Interesting experimental questions. The paper explores whether LLM-assisted agents can meaningfully contribute to causal discovery and how human feedback and retrieval impact the results.
- Has promising early results. CAST performs competitively on moderately high-dimensional datasets where traditional algorithms (e.g., FCI) become intractable.

**Weaknesses:**

(I write one bulletpoint per weakness and then detail it below)

- Writing and presentation

1. Several sections (notably the Preliminaries) are written in overly informal language and contain repeated or redundant phrasing. Citations are missing or improperly formatted (see detailed feedback below in questions/suggestions). Some notation is inconsistent e.g., between $Pa_i$ (line 93) and $Pa_G$ (line 88), or Agent $\mathcal{D}$ (line 244) vs $\mathcal{D}_{hyp}$ (line 253).
2. Definition 3 ("Causal Partitioning") originates from Zhang et al. (2022) but is presented as if novel ("A mathematically rigorous way of grouping variables is the one of Causal Partitionings, which we define below."). It also violates the disjointness property of partitions. This should be clarified explicitly. It's more of an overlapping clustering, not a partition in the strict sense.
3. Figures 1 and 2 display only three phases, although the text describes four, which is a bit confusing.



- Conceptual and methodological concerns

1. The **technical novelty** is somewhat limited. CAST mainly orchestrates existing causal discovery and RAG/HITL components rather than introducing a new algorithm or estimator. The "method" is largely a structured prompting pipeline rather than a model or optimization procedure.
2. The theoretical results hinge on ideal causal partitioning and infinite data. No identifiability or performance guarantees exist when these conditions fail: precisely the realistic case.
3. The linear workflow may be restrictive. In real expert-in-the-loop settings, humans often revise earlier causal assumptions as understanding improves. Allowing back-and-forth updates between phases (rather than a one-directional pipeline) would perhaps better reflect practice.


- Orchestration vs. Model

Performance varies drastically with the chosen LLM (e.g., Qwen 14B > o3-mini).
This suggests that CAST's success may depend more on the LLM's reasoning power than on the method itself. A useful comparison would be a single-query baseline (running the whole causal discovery pipeline with a single model query) to demonstrate the added value of the multi-agent design.


- Experimental issues

1. Scalability interpretation is overstated: claiming sub-quadratic or "$\times{}2$ scaling" from only two data points (Small->Medium) is not statistically meaningful ("This indicates that CAST scales more robustly" on line 418). More datasets or controlled scaling experiments are needed.
2. Metric choice. Reporting orientation accuracy would clarify whether CAST's improvements concern causal directionality or merely edge presence. Additionally, reporting Structural Intervention Distance (SID) would be valuable, since the future work section mentions extending CAST to more general causal reasoning tasks involving interventional queries.
3. Stability not studied. Both LLMs and human interactions are stochastic and context-sensitive. No experiment measures how variable CAST's outputs are across runs, agents, or human responses.
4. Dataset suitability bias. The authors acknowledge that CausalMan Medium contains repeated patterns easily picked up by LLMs, which likely inflates CAST's apparent scalability.
5. CausalChambers interpretation: The explanation that extra predicted edges "might reflect unmodeled physics" seems unjustified and self-serving. The ground truth is physical. Such edges are just false positives.
6. Agentic RAG performs a web search. The paper does not describe any guardrails against retrieving benchmark ground truths (e.g., filtering domains, blocking known solution pages, offline corpora, or logging/inspecting retrieved URLs). Leakage seems possible here (and RAG seems important in the ablations).
7. While the experiments are interesting, they cover only three cases and do not empirically validate key claims about CAST's internal behavior. For example, there are no checks that agents successfully "translate human-provided information into the language of causality" or that the Divide phase produces causally consistent partitions.
8. Similarly, the frequently stated advantage that CAST "does not require tedious manual specification" is not demonstrated, as all experiments were conducted by the authors themselves. A human-grounded study involving users with varying levels of causal expertise would help substantiate this claim.


- Ethical and safety aspects.

Even though the ethical statement is optional, its absence in this paper is notable.
It's important to warn users about the risks of using public agents. Such agents can store/steal data. They can also be unfair and discriminatory (as is the internet sometimes), etc.


- Missing limitations (some mentioned already in other weaknesses)

CAST provides no guarantees or identifiability results when the Divide phase is imperfect, which could make the approach unreliable for sensitive applications. Moreover, the authors acknowledge that the evaluated datasets are particularly well-suited to the Divide-and-Conquer assumption, so performance might degrade on less structured data. Finally, no stability study was conducted to assess robustness to stochasticity in LLM outputs or human inputs.

**Questions:**

- Could RAG "leak" the exact target graph from the web? Did you prevent that?

- Did you log all agent-human interactions and retrieved documents? Will these be released for reproducibility?

- Can humans revise earlier decisions (e.g., change partitioning after seeing partial results)? If not, could CAST support iterative feedback loops rather than a linear pipeline?

- Why are only additional edges inserted rather than also removing edges inconsistent with the data? Is this due to relaxing faithfulness assumptions? Wouldn't this risk over-connectivity? (c.f. CausalChambers)

- We can use HITL for the divide phase, but it seems very difficult for a human to do such a task. How feasible is this in practice?

- Have you run any human-grounded experiments with real users of varying causal expertise to test whether CAST indeed avoids tedious manual specification?

- Add an ethics statement addressing data leakage, bias in retrieved content, and human accountability when CAST assists in high-stakes domains.

- Does "on demand" mean triggered by the agent or the human? Can it be parameterized (e.g., frequency of HITL queries)?


- Writing issues

1. Figures 1-2: The diagrams currently show only three phases, while the method actually consists of four (Explain, Divide, Conquer, Combine). The Explain phase is mentioned for the first time on line 199.
2. Subsection numbering: There is a 2.1 without a 2.2 (Line 157).
3. Figure 1: Not referenced in the text.
4. Table 2 (precision column): The bolded value should be for LLM-BFS, not CAST, since LLM-BFS achieves the best precision.
5. Line 32: "Existing approaches" lacks a supporting citation.
6. Lines 38-39: Citations should appear in parentheses.
7. Line 98-99: Add a reference for MAG (Maximal Ancestral Graphs).
8. Line 183-184: Missing parentheses around citations.
9. Line 216: A citation is needed to support the statement.
10. Line 488: The author name "adamos hadjivasiliou" is entirely lowercase in the reference, capitalize properly.
11. The contributions paragraph in the introduction repeats content from the introduction itself. Consider combining them.
12. The paragraph on causal identifiability (Lines 108-116) is unclear and overly informal. Clarify wording and provide examples for "certain sets of mathematical examples" (Line 108).
13. Phrases such as "often not possible" and "often result" (Lines 111, 115) should either include supporting citations or be replaced with neutral statements like "cannot be" / "can result."
14. Line 360: The word "are" is missing.
15. Line 415: "Linked to the sparsity" is vague-clarify whether the issue is too sparse or not sparse enough.
16. Line 430: The notation $O(D^2)$ is unclear. Define $D$
17. There are two notations for parents: $Pa$ and $Pa^G$, but G is never introduced properly.
18. In the Divide phase, parameters appear inconsistently. Line 247: first occurrence of $p_{div.hyp}$, but only $p_{div}$ was defined. Line 252: first occurrence of $p_{div.critic}$, likewise undefined earlier.
19. $\mathcal{D}$ vs $\mathcal{D}_{hyp}$: The same symbol is used for both the dataset and the Divide-hypothesis agent. Please clarify.
20. Line 268: Notation inconsistency between $H$ and $C_{hyp}$.
Theorem 1: Relies on the infinite-data assumption; state this explicitly.
In the appendix, include at least one full dialogue between CAST and a human to illustrate the interaction process.

---

> ### Author Response · Authors · 2025-11-23
> **Answers and Clarifications (Part 1 of 2)**
>
> Dear reviewer WiZQ,
>
> Thanks for the very detailed feedback, and acknowledging that our method is promising. First of all, many thanks for pointing to all typos and notation inconsistencies with such detail. We integrated them in the updated manuscript. We coloured your suggested changes in $\color{blue}{blue}$. Feedback shared by multiple reviewers in $\color{violet}{\text{violet}}$.
>
> ## Concept and Methodology
> - **[Contribution and Vision]** In our vision, we seek for interactively building causal models. Therefore, we focus our contribution on the interactive HITL approach to causal structure learning, and on how that is used to tackle real-world large-scale causality. Due to known limitations when working only on observational data, those human priors are crucial. Prior work, in comparison to us, incorporates expert knowledge by either requiring structured inputs (e.g., explicit edge-level constraints, templates, or user-defined parametric priors) or assuming access to a lot of precise prior information. In CAST, and also many in real-world settings, experts interact in limited amount and with an high-level/abstract language. Therefore, **in this paper we propose a methodology to build large-scale causal models using natural language, and all the steps that we adopted serve this task and are needed to keep interaction scalable (nr.Interactions should not grow exponentially with the size of the graph)**. This is done by balancing RAG with HITL calls, while keeping it tractable by using Divide&Conquer. Our method is more than just the combination of its components because only all stages of CAST as a whole can enable that, and our contribution stands on the collective behavior of the whole system.
> - **[Theoretical Contribution]** We remark that a theoretical analysis is present also in the case of incorrect causal partitionings. Theorem 2 bounds the SHD under incorrect causal partitionings. The lack of strong identifiability guarantees is due to the assumption of causal insufficiency,  nonlinearity and nonadditive noise models. That is the norm in real-world data. To the best of our knowledge, under those assumptions, such guarantees also do not hold in any currently available method, which is why we focused our analysis on concrete error bounds for SHD instead.
> - **[Asymptotic Guarantees]** We added an additional and more explicit remark after the theorem 1 that the result is asymptotic (in line with other works).
> - **[Orchestration vs. Model]** The single-agent baselines are the LLM-BFS and Pairwise. Doing Causal Discovery on large-graphs using a single-query is not feasible due to context-window limitations.
>
> ## Experimental Setting
> - **[Performance Gap due to Skeleton or Edge diretionality]** On CausalChambers (See the table in our rebuttal to Reviewer 8q3g), we see that CAST is significantly better in terms of skeleton discovery, which is supported by its strongly superior $F_1$ and Precision. That is not as good for edge directionality, as shown for SHD (Since Precision and recall are superior, inferior SHD can only be due to the penalization derived from wrong directions).
> - **[SID]** Evaluating the structural intenventional distance is computationally unfeasible on high-dimensional data.
> - **[Stability]** We repeat all experiments for 5 random seeds [4,6,42,66,90], and we report mean and std. for every metric.
> - **[Dataset suitability bias]** Once a repeated causal structure is detected, for example by observing some patterns in the variable naming, we expect the LLM to capture and leverage that. That is one of the advantages of using LLMs that we intentionally seek. Data-driven baselines cannot leverage patterns in the meta-data, whereas CAST can.
>
> ## Ethical Aspects
> We fully agree and added an ethics and reproducibility statement discussing good practices for multiagent LLM-based systems, with special focus on the human accountability necessary in interactive settings.

---

> > ### Author Response · Authors · 2025-11-23
> > **Answers and Clarifications (Part 2 of 2)**
> >
> > ## Answers
> > - **[Retrieval of Prior Knowledge using Web-Search]** We intentionally gave CAST the capacity to gather all necessary prior knowledge from varied sources (as a surrogate-expert), and recover missing details from the provided data. Large-scale causality in real-world settings necessitates large amounts of prior knowledge due to known theoretical limits.
> > - **[Revision of Earlier decisions]** In the current version of CAST, there is not (yet) the possibility to "trace-back", and leave this for future work. Still, we agree that it could give stronger flexibility to the system and could provide further support to domain experts.
> > - **[HITL During Divide-Phase]** The Human interaction during the Divide phase can query the user for confirming whether a group of variables makes sense conceptually, for example when gathering all variables related to a specific machine in a manufacturing line. So, the interaction is kept natural and high-level.
> > - **[HITL Experiments]** We benchmarked on CausalMan with real manufacturing experts. Gathering multiple cohorts of increasing causal knowledge is an interesting experiment, but that would require a massive amount of resources, since the number of participants needed to make such experiment statistically significant would be very large.
> > - **[On-Demand Interaction]** Agents have the possibility to make tool-calls to a "RAG" and a "Human-In-The-Loop" tool. Therefore it's to the agents to decide.
> >
> > We thank again the reviewer, and hope that this rebuttal will give clarity to our work. In case additional points arise or have to be addressed more deeply, please continue this discussion.

---

> > > ### Comment · Reviewer_WiZQ · 2025-11-26
> > >
> > > My overall comment is that the rebuttal responds to most of the critiques but doesn't convincingly answer/address a lot of them.
> > >
> > > > Orchestration vs model
> > >
> > > The rebuttal mentions that the single-agent baselines are LLM-BFS and Pairwise. But these are not fair comparisons because they rely on different models than those used by the best CAST agents. Specifically, in table 1 you can see that LLM-BFS (o3-mini) actually does better$^1$ to CAST (o3-mini), while CAST (Qwen3-14B) does the best. In the paper, the authors show that performance strongly depends on the LLM (Figure 4, and table 1). This model mismatch makes it unclear how much of the improvement comes from the architecture/orchestration vs using an LLM that can handle the task better.
> > > Further, when testing against large graphs: the context-window limitation is a fair comment, but it doesn't prevent constructing a minimal-orchestration baseline to isolate the effect/impact of the authors' multi-agent design (which is their central contribution).
> > >
> > > $^1$ or comparably, depending on what you care about
> > >
> > > > Conceptual and methodological concerns
> > >
> > > The authors reiterate their motivation, but that doesn't change the initial comment, which questions novelty. Something can be valuable without being novel.
> > >
> > > Further on the practical relevance of the theoretical guarantees: the critique was that the theoretical analysis offers no meaningful guarantees in the realistic/non-ideal setting. The authors mention "no method has guarantees", but that doesn't resolve that the theoretical section doesn't give a lot of insight. This is a purely practical paper, if the theory doesn't hold in practice, its usefulness is very limited.
> > >
> > >
> > > > Writing issues and ethics statement
> > >
> > > The new revision has addressed my concerns on writing issues and now has an ethics statement.
> > >
> > > > Potential RAG leakage
> > >
> > > The rebuttal does not address the actual question, which was about leakage. The paper does not describe any safeguards preventing the RAG component from retrieving benchmark ground-truth graphs or solutions from the internet. Since RAG contributes substantially to performance in the ablations, the absence of any "contamination control" makes it unclear how much CAST genuinely reconstructs the graphs or retrieves them. The rebuttal's statement that CAST is "intentionally" allowed to gather prior knowledge does not answer the concern.
> > >
> > > This is why in one of the comments, I asked to include at least one complete dialogue between CAST and a human in an appendix to illustrate the interaction process. Even better would be to provide the logs of all agent-human interactions and retrieved documents in the supplemental material (mentioned in another comment). I understand these logs might not exist and it might be expensive to repeat the experiments, but even including a very small number of examples would increase transparency a lot and help address the leakage concern. Since the authors re-ran the experiments for five seeds during the rebuttal (mentioned above on [Stability]), including the logs for at least one of those runs would have been feasible and would substantially strengthen the paper.
> > >
> > >
> > > > Experimental setting
> > >
> > > Some concerns were addressed better than others in the rebuttal. The scalability interpretation remains unaddressed: drawing conclusions about "robust scaling" from only two data points is not statistically meaningful. On metrics, there was no reply regarding orientation accuracy, which would clarify whether CAST's gains come from directionality or skeleton recovery. Stability was partially addressed through 5 random seeds, but this does not cover variability arising from the human-in-the-loop aspects. There was also no discussion of leakage safeguards for the RAG component (mentioned above as well), or any validation of CAST's internal steps (e.g., whether the Divide phase produces meaningful causal partitions). Also, the claim that CAST reduces tedious manual specification is still not supported by evidence/user-study.
> > >
> > > > Closing comment
> > >
> > > I will sustain my initial score.

---

> ### Author Response · Authors · 2025-11-27
> **Additional Clarifications**
>
> Dear Reviewer WiZQ,
>
> Thanks for engaging and for the thorough answer. We appreciate the opportunity to clarify, and we address your points below.
>
> - **[Orchestration vs. Model]** There appears to be a misunderstanding regarding the results and the models used:
>     - **[Identical Models]**  We strictly controlled for the model. LLMs on the single-agent baselines are also tested for CAST (Qwen3-14B and o3-mini), aside from LLM-BFS which used only o3-mini due to context-window limits on Qwen3-14B.
>
>     - **[Performance Comparisons]** Since the LLMs are the same, we can fix them and see how CausalSteward performs better.
>         - **[LLM-Pairwise]** Even by excluding the benefits of the Human-Feedback, we can still observe how CAST(RAG) with Qwen3-14B performs uniformly better than LLM-Pairwise with the same LLM. In Table 1 we have the results. Same for o3-mini.
>         - **[LLM-BFS and Its precision]** You noticed that LLM-BFS(+o3-mini) has precision equal to 1, which is an artifact of the learned graph being extremely sparse: Recall and $F_1$ are very low for LLM-BFS. Its high precision is simply due to a failure to identify relationships: It is very conservative in inserting edges. CAST provides a balanced and superior $F_1$ score and significantly lower SHD.
>
>     - **[Minimal Orchestration Baseline]** The LLM-Pairwise baseline is the minimal orchestration baseline which represents the LLM performing the task with minimal guidance (just looping over all possible pairs) versus the complex orchestration of CAST. The results show that even with the same model, the CAST architecture adds significant improvements over the pairwise approach.
>
> - **[RAG Leakage & Release of Interaction Logs]** To address your request in a direct and transparent manner, we added to the appendix and supplementary materials two end-to-end interactions with CAST(HITL).
>
>     - **[New Appendix section H]** We have added a complete, step-by-step formatted transcript of a CAST interaction on the Earthquakes dataset.
>
>     - **[Supplementary Material]** We have included an additional full PDF export of a run on the CausalChambers dataset. We did not include it on the manuscript to keep its length reasonable.
>
>
> - **[Constructive Theory and Its Practical Usage]** Our theory is provided along with a practical discussion in the appendix. Our theoretical analysis for real-world scenarios is in Appendix A, Theorems 4 to 8. In our analysis, we derive error bounds depending on specific practical factors: The LLM capacity of retrieving edge constraints, and their partitioning accuracy. We further analyze and bound inter and infra-partition SHD. This is further analyzed quantitatively by the new ablations (Figure 5.1), where we indeed evaluate the performance *within* partitions. Additionally, our analysis of spurious adjacencies in the partitioning phase directly motivated the design of the merge agents ($\mathcal{M}_{hyp}$ and $\mathcal{M}_{critic}$), which act as the "safety net" predicted by the theory.
>
>
> - **[Experimental Setting]**
>     - **[Robust Scaling]** We are testing on 5 datasets, not only 2. **In the new manuscript we are testing on 5 datasets of increasing size**: Earthquakes (7 Variable), CausalChambers (34) variables, CausalMan Small (53 Variables), CausalMan Medium (186 Variables), and Neuropathic Pain Dataset (222 Variables). Further, aside from the illustrative Earthquakes dataset where no ground truth is available, we are quantitatively evaluating metrics on 4 of them, therefore the quantitative part of the evaluation spans from 34 to 222 variables.
>     - **[Orientation Accuracy]** We can use CausalChambers as an example. From the measured metrics in Figure 5.2, it logically follows that CausalSteward's strong point is on the skeleton discovery part.
>     - **[Stability in Human interaction and User-Study]** During different seed-runs, also HITL calls can be different. Therefore, given that Human-experts were involved in the benchmarking, that fundamentally "absorbs" the human and the systems' stochasticity.
>
> We thank you again for the constructive feedback. We addressed your points in the most direct way possible (like the inclusion of the full transcripts in Appendix H, clarification regarding the 5 datasets used, and the analysis of the baseline metrics). We hope that these clarifications were satisfactory.
>
> Best regards,
> The Authors

---

### Meta-Review · Area_Chair_cMtK · 2026-01-05

**Summary:**

I was put in charge of judging this paper (as an AC) late in the process. However, I spent some time reading the paper.
The main concern of the reviewers is lack of sufficient novelty.  I concur with the reviewers.  In terms of theory, the consistency results under assumption of Theorems is expected (in general consistency results in statistics are not very hard to prove).  However, I cannot imagine any reasons that these assumptions hold up on applications of LLM. This means that the theoretical results, although mathematically new, have little bearing on the practice of the proposed algorithm.  The implementation  pipeline and the idea of querying humans by the LLM is interesting. That said in general, human in the loop ideas have become recently popular . It is true and noteworthy that your algorithm requires lighter amount of input from experts. However, this novelty is not strong enough to justify acceptance to as competitive of a conference as ICLR.

**Reviewer Concerns:**

Lack of Sufficient Novelty is the main concern of the reviewers.

**Reviewer Scores:**

All reviewers recommend rejection (4,4,4)

---

### Decision · Program_Chairs · 2026-01-26

Reject